# Functional screening in human HSPCs identifies optimized protein-based enhancers of Homology Directed Repair

Juan A. Perez-Bermejo [1], Oghene Efagene[1], William M. Matern [1], Jeffrey K. Holden[1], Shaheen Kabir[1], Glen M. Chew [1], Gaia Andreoletti [1], Eniola Catton[1], Craig L. Ennis[1], Angelica Garcia[1], Trevor L. Gerstenberg[1], Kaisle A. Hill[1], Aayami Jain[1], Kristina Krassovsky[1], Cassandra D. Lalisan[1], Daniel Lord[1], B. Joy Quejarro[1], Jade Sales-Lee[1], Meet Shah[1], Brian J. Silva [1], Jason Skowronski[1], Yuri G. Strukov[1], Joshua Thomas[1], Michael Veraz[1], Twaritha Vijay[1], Kirby A. Wallace[1], Yue Yuan [1], Jane L. Grogan [1], Beeke Wienert[1], Premanjali Lahiri[1], Sebastian Treusch[1], Daniel P. Dever[1], Vanessa B. Soros[1], James R. Partridge[1] & Kristen L. Seim [1] ✉

Homology Directed Repair (HDR) enables precise genome editing, but the implementation of HDR-based therapies is hindered by limited efficiency in comparison to methods that exploit alternative DNA repair routes, such as Non-Homologous End Joining (NHEJ). In this study, we develop a functional, pooled screening platform to identify protein-based reagents that improve HDR in human hematopoietic stem and progenitor cells (HSPCs). We leverage this screening platform to explore sequence diversity at the binding interface of the NHEJ inhibitor i53 and its target, 53BP1, identifying optimized variants that enable new intermolecular bonds and robustly increase HDR. We show that these variants specifically reduce insertion-deletion outcomes without increasing off-target editing, synergize with a DNAPK inhibitor molecule, and can be applied at manufacturing scale to increase the fraction of cells bearing repaired alleles. This screening platform can enable the discovery of future gene editing reagents that improve HDR outcomes.

The discovery of CRISPR-Cas9 has transformed the landscape of gene therapy[1,2]. Ex vivo genetic editing of a patient's hematopoietic stem and progenitor cells (HSPCs) has the potential to cure a wide range of diseases, such as hemoglobinopathies and primary immunodeficiencies[3–5]. The premise of CRISPR-Cas9 gene therapy relies on the introduction of a targeted DNA double-strand break (DSB) and subsequent repair by innate cellular pathways. One pathway, homology-directed repair (HDR), utilizes DNA donor templates to enable precise gene correction, resulting in a scar-free conversion of alleles from disease-causing to non-pathogenic[6].

Despite the therapeutic potential of HDR-based ex vivo gene therapy, its clinical implementation remains a challenge for applications requiring high efficiencies of gene correction[1,7,8]. Though targeted DSBs in HSPCs can be efficiently engineered ex vivo, template-guided HDR repair of DSBs competes with the non-homologous end joining (NHEJ) and microhomology-mediated end joining (MMEJ) pathways[6,8]. NHEJ and MMEJ yield a range of sequence insertions or deletions (indels) at the break site and can lead to gene disruption. Moreover, it has been observed in HSPC xenograft models that HPSC subpopulations with long term engraftment potential preferentially

[1]Graphite Bio, South San Francisco, CA, USA. ✉e-mail: scicomms@graphitebio.com

utilize indel-producing pathways over HDR[9]. This can result in an enrichment of indels post-transplantation and a reduction of overall rates of HDR correction in vivo, potentially compromising the therapeutic potential of edited cells post engraftment. One way to mitigate this risk is to increase overall HDR correction levels ex vivo by increasing the amount of DNA donor provided. Unfortunately, excess AAV6 or ssODN, the preferred DNA templates for HDR in HSPCs, can further induce DNA damage response (DDR) pathways and limit proliferation, yield, and engraftment potential of edited cell pools[10–12]. In order to improve the safety and efficacy of HDR-based gene therapies, there is a growing need for methods that can increase HDR editing efficiency and/or reduce the negative impact of DNA repair templates.

An alternative way to improve HDR editing rates is through downregulation or inhibition of key factors involved in competing NHEJ and MMEJ pathways, such as 53BP1 and DNAPKcs (NHEJ) or DNA polymerase theta (MMEJ)[8,13–17]. However, the efficacy and safety profiles of existing DNA repair-modulating approaches in HSPCs are currently poorly understood. To increase HDR repair outcomes reliably and safely in HSPCs, we sought to systematically identify, optimize, and characterize antagonists of DNA repair pathways in ex vivo editing platforms using highly selective, target-specific, and transient protein-based inhibitors that can be co-delivered into cells with the RNP.

In this study, we describe the design and implementation of a pooled screen to identify protein variants with optimized HDR boosting capabilities. Notably, this screen is performed in primary HSPCs and utilizes HDR as a readout, enabling direct interrogation of DNA repair modulating proteins in a functionally and physiologically relevant context. We leveraged this screen to improve the potency and robustness of i53, an engineered ubiquitin variant[13,15] that increases HDR-based editing outcomes[18] by inhibiting the recruitment to DSBs of 53BP1, a factor that inhibits end-resection at DSBs[19] (an early step in the repair of breaks by homologous recombination or MMEJ). Using targeted saturation mutagenesis at the i53:53BP1 binding interface and a screening process in primary human HSPCs (a cell type with intact DNA repair, DNA damage sensing responses and other cell stress sensing signaling pathways), we identified i53 variants that introduce stabilizing interactions with 53BP1 and are more potent than i53 at increasing HDR editing outcomes in HSPCs. We also provide an extensive characterization of the resulting gene editing and cell health outcomes in HSPCs using differing levels of repair template and compare the results to a separate family of HDR enhancing small molecules. Overall, our results demonstrate the utility of our pooled functional screening system to identify promising protein-based reagents for the development of safer, more efficient HDR-based gene therapies.

## Results

### Functional screening in HSPCs identifies candidate HDR-enhancing proteins

To identify protein-based reagents that enhance HDR (Supplementary Fig. S1.1), we developed a pooled screen platform in HSPCs that uses HDR as a direct functional readout (Fig. 1A, Supplementary Fig. S1.2). In this system, HSPCs are first transduced with lentiviral libraries containing candidate protein inhibitor cDNA sequences, then edited using CRISPR-Cas9 RNPs and an AAV6 DNA donor to introduce a GFP expression cassette (UbC-GFP) via HDR at the desired locus. The GFP positive (GFP+) HSPCs fraction (i.e., HDR-positive) and the GFP negative (GFP-) HSPC fractions (i.e. HDR-negative) are isolated via fluorescence activated cell sorting (FACS) and protein variants contained within each pool are identified via next-generation sequencing.

After the development and validation of the screening system, we leveraged this platform to identify i53 variants that display improved HDR enhancing capabilities when targeting the *HBB* locus, a genetic site of therapeutic relevance for the treatment of multiple hemoglobinopathies such as sickle cell anemia and β-thalassemia[20–22]. We first screened individual and combinatorial saturation mutagenesis

libraries targeting two adjacent i53 residues L67 and H68, located at the binding interface of i53 and 53BP1 (Fig. 1B), and identified amino acid variants that were enriched in the GFP+ population relative to i53, in particular variants that introduce a positive charge at residue 67 (Fig. 1C, Supplementary Fig. S1.3). We then iterated on the sequences of two top hits from these libraries (L67R and L67H) by varying the amino acids T12 and T14 simultaneously, which revealed changes that could further increase the HDR-based readout (Fig. 1D, Supplementary Fig. S1.4). These included the introduction of negative charge or, in the case of parent L67H, a positively charged histidine at position 14 (Fig. 1E). In all libraries tested, consistent enrichment results were observed across degenerate codons and amino acids with similar properties. Promising variants from each screen were validated by editing HSPCs transduced using smaller pooled libraries and/or individual variant transductions (Supplementary Fig. S1.5) using the HBB-UbC-GFP AAV6 donor.

Improved i53 variants identified in each screen were recombinantly produced for biophysical characterization and assessment as purified protein-based gene editing reagents. Their binding to the 53BP1 Tudor domain were evaluated using size exclusion chromatography, biolayer interferometry, and TR-FRET. All recombinant variants complexed with the 53BP1 Tudor domain and exhibited increased binding affinity relative to parental i53 (Supplementary Fig. S1.6, Supplementary Table S1.1), suggesting a correlation between the improved affinity of the selected variants and their enrichment in the GFP+ population in our screening platform. Taking this data altogether, we chose four i53 protein variants that represent the sequence diversity observed among top performing hits (L67H, L67R, T12V.T14H.L67H, T12Y.T14E.L67R) to further characterize as protein-based HDR enhancers in HSPCs.

### Identified i53 variants display additional intermolecular interactions with 53BP1

To understand the molecular basis of the improved activity and binding of the de novo i53 variants identified in our screens, we solved crystal structures of each variant bound to 53BP1 (Fig. 2, Supplementary Figs. S1.1, S2.2, S2.3, S2.4, S2.5 and S2.6, Supplementary Table S2.1, Supplementary Data 1 and 2). We observed that the L67R change enables a new 2.8 Å H-bond with D1550 of 53BP1, and that L67H forms a network of inter- and intramolecular H-bonds, including a water mediated H-bond to S1554 of 53BP1 and an H-bond to D64 on i53. The intermolecular H-bond to D64 could help lock the i53 loop conformation observed in the crystal structure and lower the entropic penalty for binding. The new interactions identified at position 67 were also present in the crystal structure for T12V.T14H.L67H and T12Y.T14E.L67R. For T12V.T14H.L67H, additional stabilizing interactions were observed at positions T12V and T14H: a water mediated bridge between T14H and Y1502 and Van der Waals contacts between T12V and 53BP1. Interestingly, the T12V mutation alters the polarity of the interface, resulting in displacement of a water molecule and formation of an intermolecular H-bond between D1521 and Y1523 in 53BP1. For the T12Y.T14E.L67R structure, T14E forms a 2.6 Å H-bond to Y1502 and a 2.9 Å H-bond to the backbone amide of Y1500. The T12Y mutation forms additional Van der Waals contacts at the protein-protein interface. These analyses demonstrate that each amino acid change in the variants selected from our screen form stabilizing interactions at the 53BP1 interface, resulting in tighter binding and validating the use of this screening approach to identify gain of function protein variants.

### 53BP1 antagonists increase HDR outcomes by reducing NHEJ directed repair

Top i53 protein variants from each round of screening validation were next assessed as HDR-boosting additives for ex vivo editing of HSPCs by adding the purified protein directly to the electroporation buffer

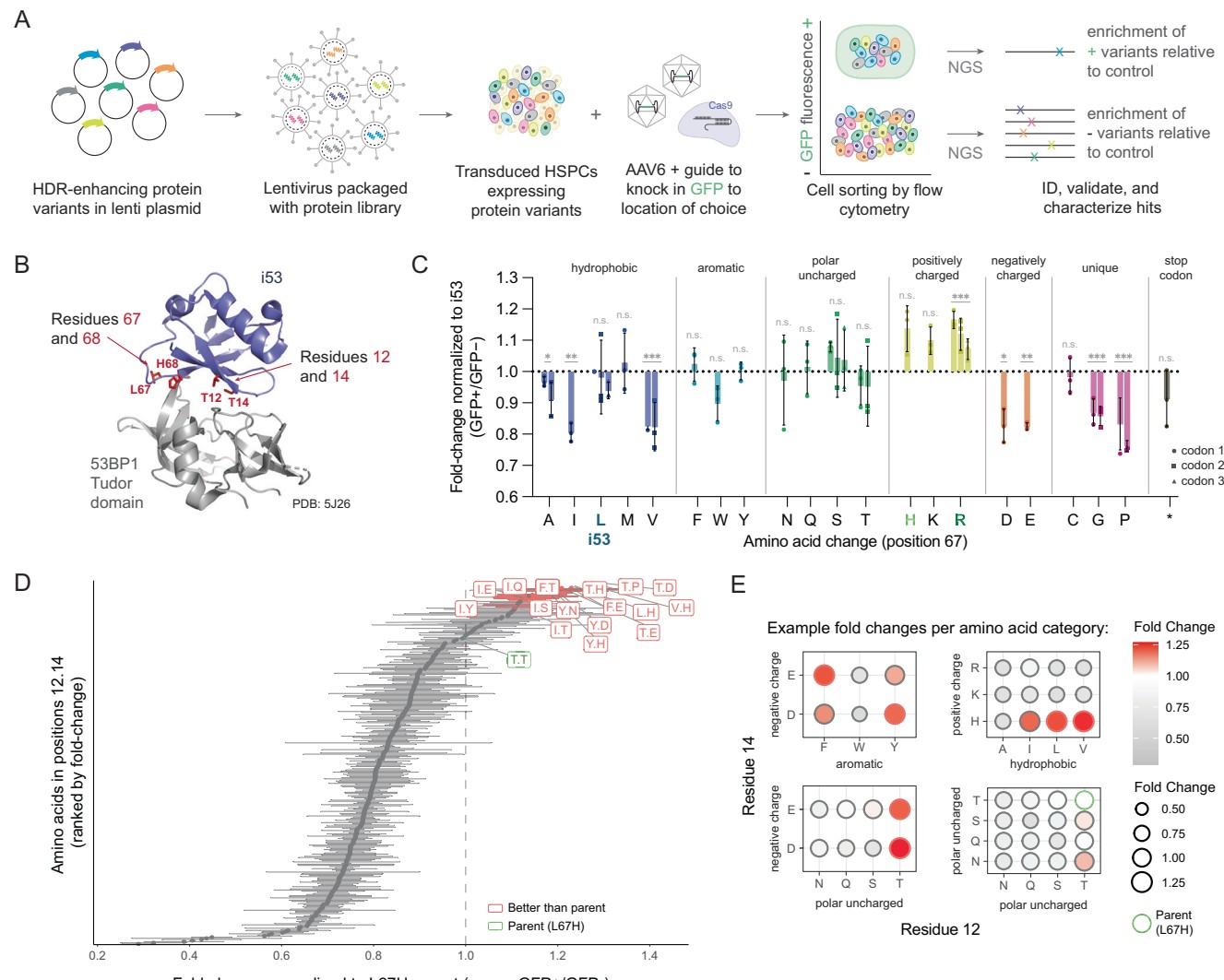

**Fig. 1 | A lentiviral-based pooled screening platform in HSPCs identifies HDR-enhancing variants of i53. A** Schematic outlining the lentiviral-based pooled screening platform in HSPCs used to identify protein-based additives to increase HDR at a Cas9-mediated cut site of interest using an AAV6 DNA donor template. Protein variants are encoded in lentiviral libraries; once integrated into the genome the sequences can be amplified from HSPC subpopulations. To functionally quantify homology-based repair in pooled libraries, transduced cells are edited using Cas9 RNP and an AAV6 template that encodes for a GFP insertion at the cut site of interest (e.g. *HBB* gene). Post editing (3–5 days), cells are sorted via flow cytometry into GFP+ and GFP- populations. Genomic DNA is extracted from each sorted cell population, sequenced via NGS, and analyzed to determine the distribution of variants relative to a control. **B** Residues targeted for mutagenesis were chosen by their proximity to the binding interface between with 53BP1 and i53 and are shown in red (T12, T14, L67, H68). **C** Example enrichment of residues following screening with a saturation mutagenesis (NNK) library at position L67 (parent: i53, library size = 32 unique codons encoding 20 variants). *n* = 3 separate pooled

analyses and mean ± SD depicted. Each bar represents a unique codon for that amino acid. n.s. not significant; *p < 0.05; **p < 0.01; ***p < 0.001. Two-tailed *t*-test with Holm-Šídák correction for multiple comparisons. Exact *p*-values are reported on Source Data file. Of the 19 new variants tested, one (L67R) was found to be significantly enriched relative to parent i53 (L67), although L67H was borderline significant and was also moved on to subsequent validation. **D** Example enrichment of residues following a combinatorial library at positions T12 and T14 (parent: L67H), library size = 324 variants for which all replicates were enriched over parent are highlighted in red (16 variants, or 5%). *n* = 3 separate pooled analyses; bars represent mean ± SEM. Selected top hits were subsequently validated in focused libraries and experiments with purified recombinant protein. **E** Dot plot representation of variant fold change enrichment in combinatorial analysis, clustered by amino acid properties. Variations of residues 12 and 14 shown on the x-axis and y-axis, respectively. Additional information for this library shown in Supplementary Fig. 1.4. **C–E** Source data are provided as a Source Data file.

containing Cas9-RNP (Supplementary Fig. S3.1A, S3.2). First, variants along with parental (WT) i53 were tested for HDR-mediated insertion of a fluorescent reporter (Supplementary Fig. S3.1B) at four clinically relevant loci: *HBB*[20], *HBA*[23], *CCR5*[24] and *IL2RG*[25]. The selected i53 variants outperformed WT i53 at all loci, leading to 1.5–2.5 x increase in the HDR fraction over the no protein control (Fig. 3A, Supplementary Fig. S3.3). To further evaluate the i53 variants in a clinically relevant gene editing strategy, they were used for HSPC editing using an AAV6 donor designed to correct the sickle cell disease causing polymorphism in the *HBB* gene (HBB-SNP AAV6)[20,21,26,27] followed by direct

amplicon sequencing as a readout (Supplementary Fig. S3.1C). Compared to the GFP knock-in reporter system, this assay provides a related but more clinically relevant and comprehensive readout of editing outcomes in HSPCs. Incorporating the i53 variants resulted in a significant increase in the proportion of HDR-corrected alleles relative to WT i53 at two different protein concentrations (Fig. 3B, Supplementary Fig. S3.4). Next, we quantified the potency improvement by testing a range of concentrations for editing HSPC cells and found the variants induce a ~3-fold improvement in potency compared to parental i53 protein (Fig. 3C, Supplementary Fig. S3.5). The inclusion of

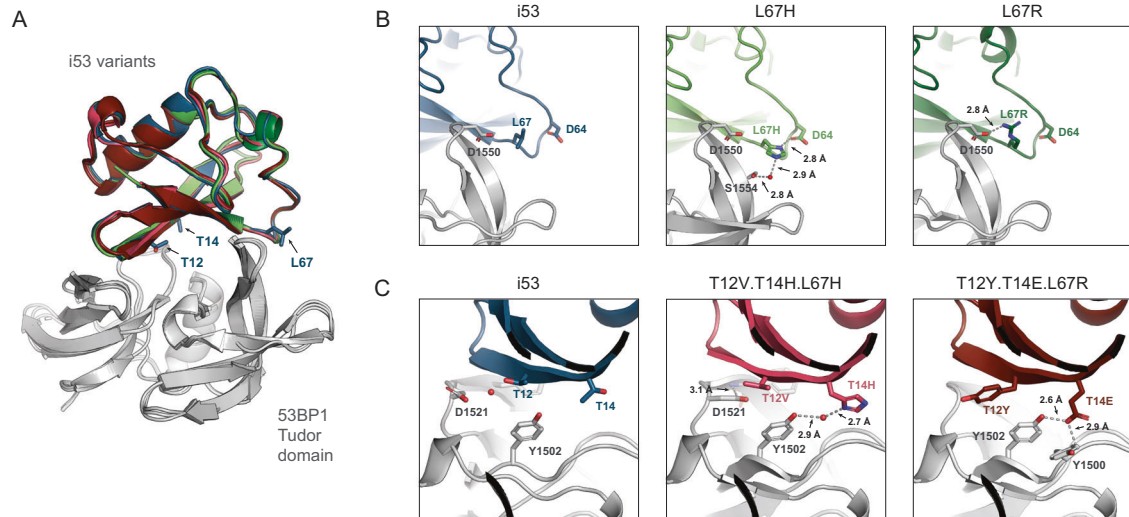

**Fig. 2 | i53 variants display additional molecular interactions with 53BP1.** Crystal structures of i53 variants bound to 53BP1 Tudor domain. **A** Structural alignments of 53BP1 Tudor domain (yellow) bound to i53 variants: WT (blue, PDB: 8SVG), L67H (light green, PDB: 8SVI), L67R (dark green, PDB: 8SVH), T12V.T14H.L67H (light red, PDB: 8SVJ), and T12Y.T14E.L67R (dark red, PDB: 8T2D).

**B** Zoomed in view of the complex at the solvent-exposed loop preceding β5 for i53, L67H, and L67R. **C** Zoomed in view of the complex near i53 residues 12 and 14 for wild-type i53, T12V.T14H.L67H, and T12Y.T14E.L67R. Hydrogen bonds are denoted by black dotted lines.

the i53 variants did not result in an increase in the fraction of apoptotic or necrotic cells at 72 h post-editing (Supplementary Fig. S3.5).

To further elucidate the effects of HDR enhancement in the profile of other edits, we characterized how the optimized i53 variants impact DNA repair outcomes in HSPCs. A detailed analysis of amplicon sequencing reads of the *HBB* locus after editing allowed us to classify the four main outcomes as follows: (1) the desired HDR event mediated by the AAV6 template ("HDR"), (2) unedited alleles ("WT"), (3) HDR mediated by the highly homologous delta-globin gene ("HBD"), and (4) indels of various lengths. To distinguish whether indels resulted from MMEJ or NHEJ pathways, we used a knockdown of an enzyme known to mediate MMEJ, DNA polymerase Theta (POLQ)[28,29], and were able to further classify indels into POLQ-dependent ("MMEJ") and POLQ-independent edits (non-homologous end joining; "NHEJ") (Supplementary Fig. S3.6). Using these classifications (Supplementary Data 3 and 4), we determined that the addition of i53 variants results in an increase of HDR by specifically reducing NHEJ events but not MMEJ nor other types of outcomes, consistent to their role as 53BP1 antagonists (Fig. 3C and 3D, Supplementary Fig. S3.5 and S3.7).

We hypothesized that HDR-enhancing molecules could allow for the reduction of DNA repair template while maintaining or improving HDR outcomes, leading to a reduction in the cytotoxicity of the editing process[8,10,30]. To test this, we edited HSPCs in the presence of i53 variants with both a high and low dose of HBB-SNP AAV6 and measured cytotoxicity by yH2AX phosphorylation and p21 expression, key markers for p53-mediated DDR[7,31]. As expected, HSPC edited with i53 variants and low viral multiplicity of infection (MOI) showed comparable HDR levels to those obtained using high dose AAV6 and no additive molecule. However, we also found that the low MOI condition displayed a significant reduction in DDR marker expression (Fig. 3E, Supplementary Fig. S3.8). Interestingly, we observed that the increase in HDR associated with a higher AAV6 dose was mediated by reducing MMEJ and HBD outcomes, while NHEJ edits were unaffected (also observed in Supplementary Fig. S3.1F). This is consistent with the observation by others that HDR displaces MMEJ indels when the homology donor is present[32]. In contrast, i53 variants increase HDR by only reducing NHEJ events (Fig. 3D, Supplementary Fig. S3.8). Together, these results demonstrate that i53 variants specifically replace NHEJ events with HDR outcomes and can be used in combination with

lower levels of AAV6 donor DNA to enhance HDR in HSPCs and mitigate DDR associated with the DNA repair template.

## DNAPKcs inhibitors and i53 variants can act synergistically to improve HDR

Next, we studied how the activity of i53 variants compares with established NHEJ inhibitors. Small molecule inhibitors of the catalytic subunit of DNA-PK (DNAPKcs), an enzyme involved in NHEJ repair, have been previously described to increase HDR outcomes in gene editing[16,17,33–37]. We first tested a subset of commercially available DNAPK inhibitors (DNAPKi) for HDR boosting capabilities at the *HBB* locus in HSPCs and found AZD7648[37] to be the most effective (Supplementary Fig. S4.1), as it has also been reported by recent studies[16,17]. Comparing the use of AZD7648 with the i53 variant L67R across different loci, we found that DNAPKi and L67R have locus-dependent effects on HDR outcomes, suggesting that DNAPKcs- and 53BP1-mediated NHEJ pathways are utilized differently depending on the target genomic sequence. Indeed, we found that combining both inhibitors resulted in an improvement of HDR larger than that of L67R or AZD7648 alone (Fig. 4A, Supplementary Fig. S4.2). Thorough analysis of the editing outcomes at *HBB* revealed that, although both inhibitors contribute to an overall reduction of the fraction of NHEJ indels, AZD7648 preferentially reduces −1, +1 and +2 INDELs at this locus whereas i53 variants have a larger effect on remaining edits (for example -2, -5, -7). The combination of both types of inhibitors effectively reduces NHEJ indels independent of length (Fig. 4B, Supplementary Fig. S4.3), highlighting the different but synergistic nature these inhibitors, and pointing towards an orthogonal mechanism of action.

Based on our previous observation that HDR enhancing additives can allow for a reduction of DNA template concentration while preserving HDR levels, we studied the impact of NHEJ inhibitors at varying concentrations of donor template. We found that, at the locus of study *HBB*, there was an inverse correlation between AAV6 MOI and HDR fraction enrichment (compared to no additive control), although this correlation was not significant for AZD7648 alone (Fig. 4C, Supplementary Fig. S4.4). We also observed that, in the absence of AAV template, the use of 53BP1 inhibitors still reduces NHEJ editing, correlated with the concomitant increase in the contribution of other

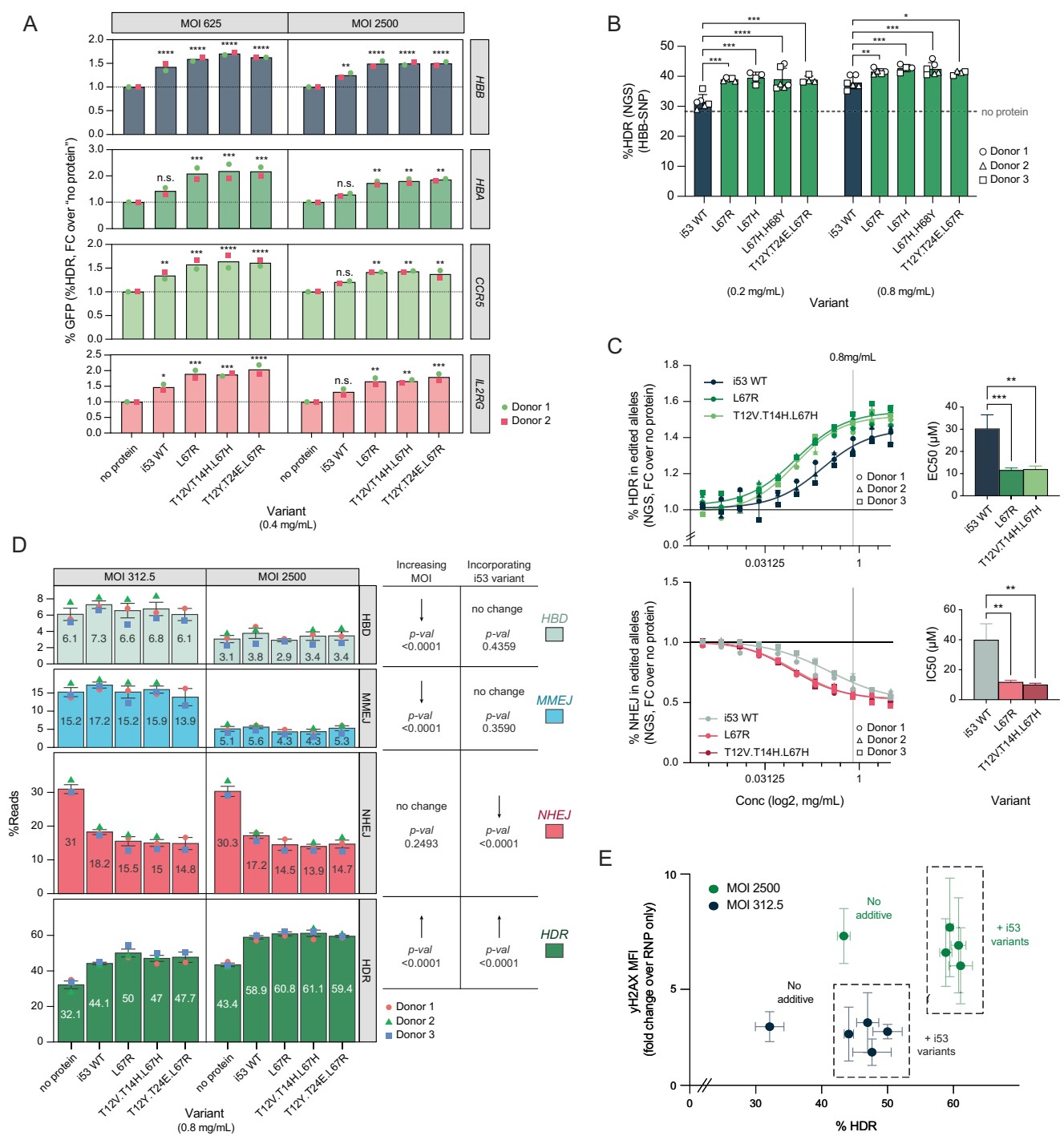

homology-based outcomes such as MMEJ or HBD (Supplementary Fig. S4.4B). Similar to i53 variant alone, use of AZD7648 alone or in combination with L67R had no significant impact on yH2AX phosphorylation, although it did decrease p21 expression (Supplementary Fig. S4.4C). Similarly, the use of either additive did not have a significant impact on cell recovery 24 h post editing, but the combination of both seemed to have a small but significant effect (Supplementary Fig. S4.4C and D). Further studies will be needed to evaluate the effect of these additives on cell yields, especially beyond the 24 h timepoint. We find, therefore, that the additives do not seem to dramatically inhibit the DDR but enable the reduction of the DDR through lowering the amount donor template needed to achieve high HDR frequencies.

To determine the safety profile of the different NHEJ inhibitors for ex vivo gene editing, we examined editing at a previously known off-

target site for the *HBB* gRNA, "OT-1", by NGS[26,38]. While the i53 variants did not impact editing at *HBB* OT-1, treatment with AZD7648 resulted in an increase in off-target indels (from 1.2% to ~2%) (Fig. 4D). This increase was caused by an increase in a −9 deletion, which also has been observed by others and could be attributable to the MMEJ repair pathway[16] (Supplementary Fig. S4.5). These results suggested that, even if DNA damage response markers are unaffected, addition of AZD7648 can result in an increase in off-target genotoxic effects.

### i53 variants increase HDR in HSPCs with surface markers of LT-HSCs without increasing genotoxicity

To evaluate the feasibility of incorporating i53-based inhibitors in a therapeutic setting, HSPC cells were edited at a larger scale (~100–200 M cells per condition), closely resembling the scale used for

**Fig. 3 | i53 variants selectively target NHEJ and enable a reduction of DNA repair template. A** Fold change in %GFP-expressing cells when purified variants of i53 are incorporated as protein-based additives to HDR-mediated GFP incorporation into HSPC at different alleles. Absolute numbers shown in Figure S3.2D. $n = 2$ from two different HSPC donors (both male, $7.5 \times 10^5$ cells/cuvette, split across 2 MOI conditions). Bars represent mean. Two-way ANOVA with Dunnett correction. n.s. = non-significant; **$p < 0.01$; ***$p < 0.001$; ****$p < 0.0001$. Exact $p$-values provided in Source Data File. **B** %HDR in HSPCs at *HBB* when edited using sickle-cell correcting HBB-SNP AAV6 with different purified variants of i53 at two different protein concentrations and an MOI of 312.5 ($5\text{-}7 \times 10^5$ cells/cuvette). For variants i53 and L67H.H68Y, $n = 6$ and for variant L67R and no protein conditions, $n = 5$ (three HPSC donors). For variants L67H and T12Y.T14E.L67R, $n = 4$ (two HSPC donors). Mean ± SD depicted. Two-way ANOVA with Dunnett correction. n.s = non-significant; **$p < 0.01$; ***$p < 0.001$; ****$p < 0.0001$. Exact $p$-values reported on Source Data file. **C** Dose response curves of i53, L67R, and T12V.T14H.L67H using HBB-SNP AAV at an MOI of 625 in HSPCs. %HDR and %NHEJ iare shown as fold change over no protein additive. Vertical dotted line: selected working concentration (0.8 mg/mL). $n = 3$ separate HSPC donors ($7.5 \times 10^5$ cells/cuvette, across 2 AAV6 conditions). Error bars: mean +/- SD. Four-parameter dose response curve fit. Bar charts represent the EC$_{50}$ and IC$_{50}$, defined as the i53 variant concentration required for a 50% of maximal increase in HDR or decrease in NHEJ, respectively, along with 95% confidence intervals (error bars). Analyzed by one-way ANOVA, with Sidak adjustment for multiple comparisons. Exact $p$-values provided in Source Data file. **D** Effect of increasing MOI and incorporating i53 variants on the editing outcomes at the HBB cut site as quantified by NGS analysis. $n = 3$ different HSPC donors ($2 \times 10^6$ cells/cuvette, across 2 MOI conditions); mean ± SEM depicted. Two-way ANOVA analysis, only main effects reported. **E** %HDR relative to induction of the DNA Damage Response (DDR) as measured by flow cytometry for phosphorylation of histone H2AX (γH2AX) in cells edited with and without addition of the i53 variants, at two different MOI. $n = 3$ different HSPC donors and mean ± SEM is depicted. i53 variants tested and grouped: i53WT, L67R, T12V.T14H.L67H, T12Y.T24E.L67R. $p$-values: MOI effect on γH2AX < 0.0001; MOI effect on HDR < 0.0001; additive effect on γH2AX < 0.3606; additive effect on HDR < 0.0001. Two-way ANOVA with Sidak adjustment. **A**–**E** Source data are provided as a Source Data file.

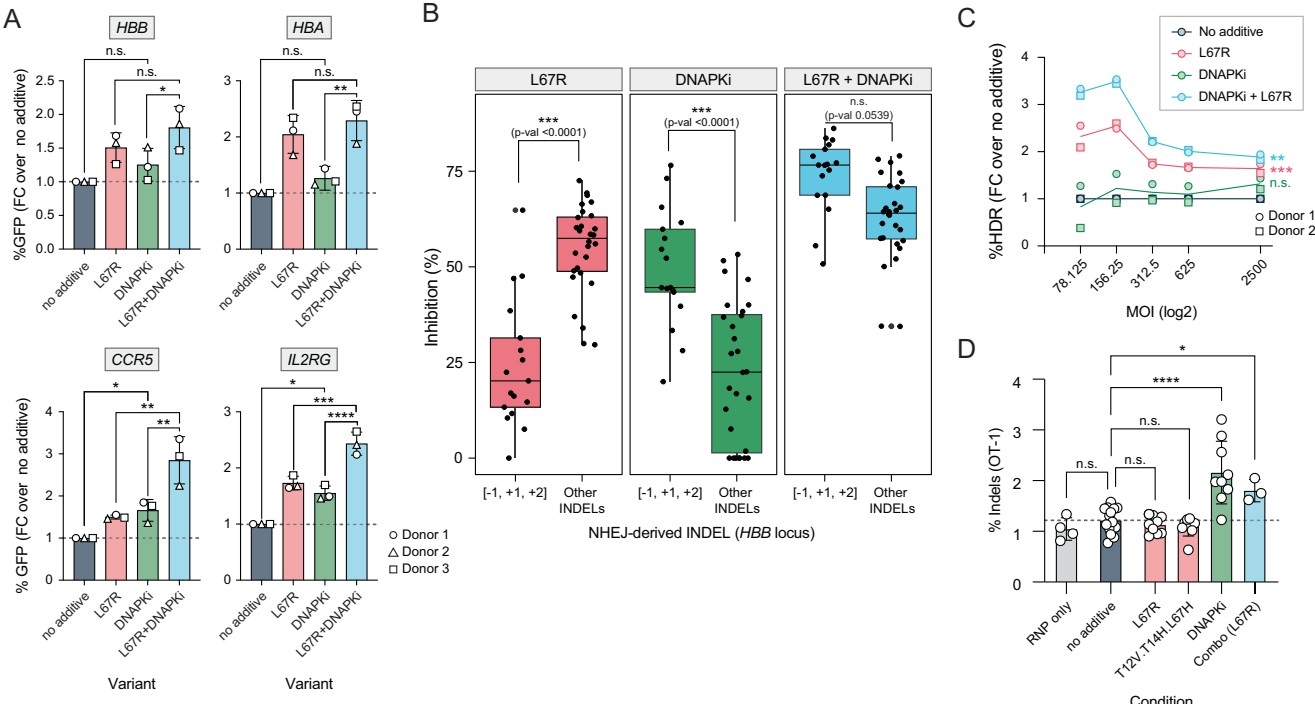

**Fig. 4 | i53 variants can be combined with DNAPKcs small molecule inhibitor AZD7648 for additional reduction in indels. A** Observed fold change in %GFP-expressing cells (%HDR) when L67R (0.8 mg/mL) is incorporated to an HSPC editing protocol for GFP knock-ins; post editing, cells were resuspended in media containing AAV targeted at *HBB, HBA, CCR5,* and *Il2RG* (MOI = 2500) with and without the addition of a DNAPKi (AZD7648, 0.5 μM). An equivalent plot with absolute % HDR numbers is shown in Supplementary Fig. S4.2. $n = 3$ different donors (all male, $1.25 \times 10^6$ cells/cuvette, split +/- DNAPKi) and mean ± SD is depicted. One-way ANOVA with Donnett adjustment for multiple comparisons. Exact $p$-values reported on Source Data File. **B** Differential effects of NHEJ inhibitors (L67R and AZD7648) alone and in combination on the %inhibition of NHEJ-derived INDELs either of length [-1, -2, +2] or other lengths, when editing the *HBB* locus. MMEJ edits are excluded. MOI = 312.5. $n = 3$ different HSPC donors. Analysis by one-way ANOVA with Donnett adjustment for multiple comparisons. n.s. = non-significant; ***$p < 0.001$. Center line denotes the median value; lower and upper hingers of box plot denote first and third quartile; whiskers extend to the smallest value at most 1.5x the inter-quantile range. **C** %HDR (NGS) in HSPCs at *HBB* when edited using sickle-cell correcting HBB-SNP AAV and L67R (0.8 mg/mL) with and without the incorporation of AZD7648 (0.5 μM), at increasing MOIs. $n = 2$ separate HSPC donors ($1.2 \times 10^7$ cells/cuvette, split across 6 MOI conditions +/- DNAPKi). Significance values are provided for slope being different from zero. n.s.: not significant ($p = 0.2717$), **$p < 0.0057$, ***$p < 0.0001$. **D** Off target indels at OT-1 in cells edited at *HBB* without NHEJ inhibitors as compared to OT-1 indels in cells edited with the addition of i53 variants (0.8 mg/mL), AZD7648 (0.5 μM) or a combination. Data is compiled from six independent experiments in which eleven different HSPC donors were used. For no protein conditions, $n = 15$ (using 11 HPSC donors across all six experiments). For conditions including variants L67R and T12V.T14H.L67H, $n = 9$ (using six HSPC donors across two experiments). For conditions using DNAPKi, $n = 9$ (using nine HSPC donors across five experiments). For the combo conditions (DNAPKi + L67R), $n = 3$ (using three HSPC donors from one experiment) and for RNP only, $n = 4$ (using four HSPC donors across two experiments). Mean ± SD depicted. Analysis by one-way ANOVA with Donnett adjustment for multiple comparisons. n.s. = on-significant; *$p < 0.05$; ****$p < 0.0001$. Exact $p$-values are reported in Source Data file. **A**–**D** Source data are provided as a Source Data file.

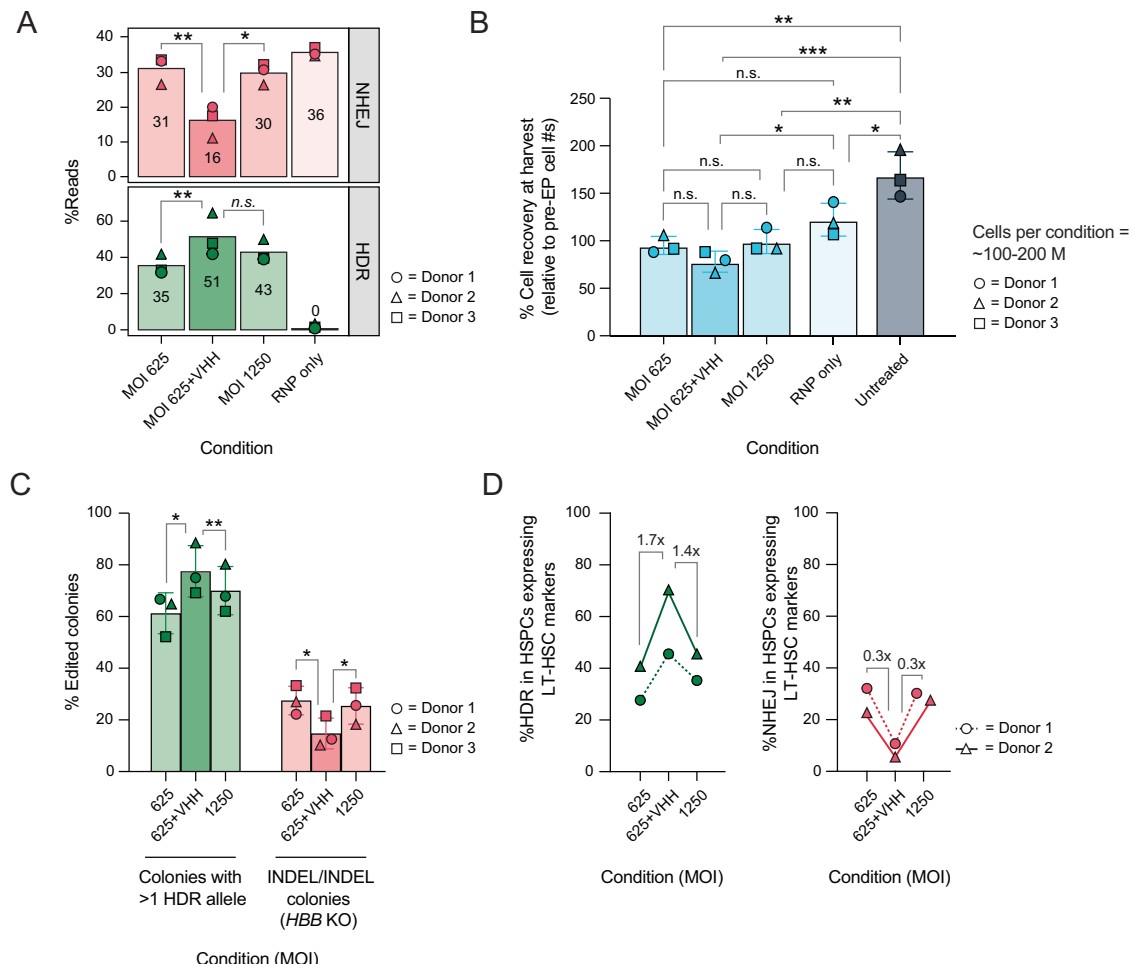

**Fig. 5 | i53 variants increase HDR in phenotypic LT-HSC subpopulations and increase the number of cells with successfully corrected alleles. A** HDR and NHEJ editing outcomes in CD34+ HSPCs that were edited with Cas9 RNP and HBB-SNP AAV6 in 3 medium scale manufacturing runs ( ~ 100-200 M HSPCs per editing condition for each donor). $n = 3$ different HSPC donors. Analysis by two-tailed paired $t$-test, with Holm-Šídák multiple comparison correction. n.s. = non-significant ($p = 0.1275$); *$p < 0.05$; **$p < 0.01$ (0.0051 in HDR panel, 0.0083 in NHEJ panel). **B** Percent cell recovery post editing (harvested 24 h post electroporation, calculated by dividing cell count at 24 h by the number of cells electroporated for each condition). Cell counts and viabilities for each condition are shown in Supplementary Fig. 5.1E and in Supplementary Table S5.1. $n = 3$ different HSPC donors

and mean ± SD depicted. Analysis by one-way ANOVA. n.s. = non-significant; *$p < 0.05$; **$p < 0.01$; ***$p < 0.001$. Exact $p$-values reported in Source Data file. **C** Percent of colonies bearing either ≥1 HDR allele (green) or 2 indel alleles (red). Individual colonies were genotyped at $HBB$ cut site locus. $n = 3$ different HSPC donors and mean ± SD depicted. Analysis by two-tailed paired $t$-test with Holm-Šídák multiple comparison correction. *$p < 0.05$; **$p < 0.01$. Exact $p$-values reported on Source Data file. **D** %HDR and %NHEJ in HSPC subpopulation expressing surface markers associated with LT-HSCs (CD34 + CD45RA-CD90 + CD201 + CD49f + CD49c + ), sorted from the bulk edited cells (donor 1 and 3 only). VHH = T12V.T14H.L67H. $n = 2$. Numbers above bars denote average fold change across two donors. **A**–**D** Source data are provided as a Source Data file.

clinical product manufacturing[27]. Consistent with previous results, we observed that adding i53 variants increased HDR and decreased NHEJ-mediated outcomes without increasing genotoxicity (off-target editing by guide-seq and OT-1 sequencing; chromosomal translocation by karyotyping; and translocation-sequencing assay) (Fig. 5A, Supplementary Fig. S5.1 and S5.2, Supplementary Tables S5.1 and S5.2). Similarly, we observed no significant difference in cell recovery at 24 h post editing (although there was a small but nonsignificant drop in recovery, driven by one biological replicate, Fig. 5B, Supplementary Fig. S5.1E, Supplementary Table S5.3), colony forming unit recovery and composition (CFU) (Supplementary Fig. S5.3) or cell type population by single-cell RNA sequencing (Supplementary Fig. S5.4; Supplementary Table S5.4) when i53 variants were included in the editing protocol.

An important metric for the ability of a genome edited HSPC cell pool to correct disease is the percent of cells (rather than alleles) that contain the desired editing outcome. It is also important to control the cells that are homozygous for detrimental outcomes such as indels,

which could result in cells with no expression of the gene of interest. To address the impact of i53 variants in the single cell genotypes, we performed amplicon sequencing on individual colonies from the CFU assay. We observed that using an i53 variant resulted in a significant ( ~30%) increase in colonies bearing at least one HDR-corrected allele, and a significant decrease ( ~50%) in cells with homozygous $HBB$ KO (indel/indel) (Fig. 5C, Supplementary Fig. S5.3).

Additionally, the success of genome edited HSPC cell therapy relies on the successful engraftment of edited cells. As such, it is important to evaluate if the HDR improvement provided by i53 variants translates to true long-term HSPCs (LT-HSC), which is the subpopulation of CD34+ with robust engraftment potential[9,36–38]. To test this, we implemented an immunophenotyping panel to sort different subpopulations from edited HSPCs from two of the donors. Amplicon sequencing revealed that i53 variants provided an increase in HDR and a reduction in NHEJ in all subpopulations analyzed, including HSPCs with surface markers associated with LT-HSCs (CD34 + CD45RA-CD90 + CD201 + CD49f + CD49c+)[9,39–41] (Fig. 5D, Supplementary Fig. S5.5).

Taken together, these results suggest that using i53-derived protein inhibitors such as the ones identified with our functional screening platform can drastically improve the fraction of HDR edited alleles in phenotypic LT-HSCs and also increase the fraction of cells carrying at least one HDR repaired allele. This improvement is provided without a detectable increase in off-target editing or other cytotoxicity events.

## Discussion

One of the most important challenges for therapeutic CRISPR-Cas9-mediated genome editing is controlling DNA repair outcomes to ensure clinically relevant levels of the desired change are introduced[1]. To identify functional protein-based additives that can modulate DNA repair and improve HDR-based outcomes in ex vivo gene therapy workflows, we developed a pooled screening platform that uses HDR as the functional readout. Unlike most screens, we performed the screen in a clinically relevant primary cell type without chromosomal abnormalities and with normal signaling pathways and DNA repair mechanisms. By performing a functional screen in HSPCs, we can identify molecules with high translational potential while reflecting cell type-specific editing outcomes and DNA repair pathway utilization[42,43]. Our strategy, which was based on screening in viable cells, also would eliminate candidates with high toxicity as those would not pass through the pooled screen against candidates that did not have toxicity. In this study, we leverage this functional screening system to assess focused libraries targeting the protein-protein interface of i53, an engineered ubiquitin derivative, and its binding target, 53BP1. By screening in this system, we facilitated the discovery and selection of tighter binding i53 protein variants that are stable, soluble, and inherently functional for boosting HDR in HSPCs. Crystal structures of i53 variants selected through each round of screening confirmed that each i53 mutation provides new non-covalent interactions with 53BP1, resulting in increased potency.

The observation that 53BP1 antagonists exhibit a different specificity towards editing outcomes compared to DNAPKcs or MMEJ inhibition confirms that multiple orthogonal, or at least partially independent mechanisms, contribute to indel outcomes[44]. Compared to recent reports using the DNAPKi small molecule AZD7648 to improve homology repair in a genome editing context[16,17], we observed a more modest increase in HDR when using either 53BP1 or DNAPKcs inhibitors alone. We hypothesize that this difference could be attributable to our report having a higher baseline HDR rate, given that we observe editing improvement become more dramatic when using lower baseline HDR rates. In addition, we also report that an increase in the concentration of homologous repair template (e.g., Adeno Associated Virus (AAV)) results in an enrichment of HDR outcomes by downregulation of alternative homology-mediated outcomes, especially MMEJ, possibly by competition with the endogenous repair templates. Taken together, these findings suggest a model where the sequence composition of the gene editing outcome can be modified in a context-specific manner (locus, homology donor, cell type) to target specific pathways and steer the gene repair machinery towards desired outcomes, enabling more precise and predictable gene editing approaches.

Our data demonstrates that editing additives that increase HDR rates also enable the reduction of template DNA required for precise editing. Typically, the repair template doses required for efficient HDR in HSPCs are inherently cytotoxic, especially for LT-HSC, compromising long term stability and potential for engraftment[10–12]. We show that using HDR boosting reagents and reducing the DNA template results in a net improvement of DDR metrics, which could improve the therapeutic potential of an edited cell pool. Furthermore, by using reduced AAV6 levels and an improved i53 variant to edit cells at clinically relevant scale, we validated that the HDR improvement persists in colony-forming cells and in cells expressing LT-HSC markers, increasing the fraction of cells bearing repaired alleles and outlining the

optimized inhibitors as promising additions to clinical ex vivo gene therapy workflows. Despite these results, additional long-term studies exploring impacts on cell health and yields beyond 24 h in culture and on the overall transplantation efficiencies and regenerative properties of edited cell pools will be required to further vet safety and applicability of these molecules for clinical ex-vivo cell and gene therapies. Future studies will also be needed to evaluate the efficacy and safety profile of these antagonists when used in genome editing of additional loci, across a broader range of HSPC donors and/or in different cell types, allowing for more generalized understanding of the implications of NHEJ, and in particular 53BP1, inhibition in therapeutic genome editing.

This study also highlights the importance of characterizing any potentially detrimental effects that could impact the prospect of therapeutic gene editing. The observation that AZD7648 significantly increased indels in the main off-target site for *HBB* editing is surprising and raises concerns about the potential for chromosomal translocations, which may limit the clinical application of DNAPK inhibitors for some targets. Future studies will be needed to investigate whether this increase is driven by the DNAPKcs inhibition or specific to the small molecule being used, and whether the increase is biologically significant. Of note, we were not able to observe any undesired effects in off-target editing, chromosomal translocation, karyotyping, or cell type composition when using the optimized 53BP1 protein antagonists for *HBB* locus editing, providing an additional level of confidence on the safety of the additives. As discussed above, additional studies using in vivo models will be needed to better evaluate long-term efficacy and safety of these reagents.

In conclusion, we have described the development of a functional screening strategy for identifying improved 53BP1 antagonists that increase HDR repair outcomes in HSPCs. This strategy was leveraged to identify i53 variants with new stabilizing interactions with 53BP1 that improve HDR fraction enrichment and decrease NHEJ. In addition, we provide an extensive characterization of the DNA repair outcomes when these optimized i53 variants and other DNA repair pathway inhibitors are used in the editing of the *HBB* locus, a clinically relevant target for gene therapy. Notably, the screening platform described in this study can be easily tailored to identify protein-based inhibitors of other DNA repair pathways (for example, MMEJ) or to find optimized reagents for cell editing in cell types other than HSPCs. This platform can also be paired with other library selection-based methods (such as phage-, ribosome-, or yeast-display) to increase throughput and enable rapid discovery of optimized gene editing reagents. This type of screening platform presents great potential to identify functionally optimized gene editing reagents that help modulate DNA repair at specific loci, expanding the therapeutic genome editing toolkit and eventually contributing to the development of safe, efficacious, and more precise gene editing therapies.

## Methods

### Molecular cloning (i53 lentiviral vectors)

The construct for the lentiviral-based expression and screening of i53 variants was cloned from a third-generation lentiviral plasmid (Lenti SFFV) purchased from Twist Biosciences. An empty vector was constructed to include BamHI and NsiI restriction enzyme cut sites upstream of a T2A-mCherry-WPRE cassette (to enable fluorescence-based monitoring of cells expressing the i53 variants). The sequences for i53 variants were either ordered as gene fragments from Twist Biosciences or IDT or amplified from previously constructed plasmids using primers designed to introduce the desired amino acid variation(s). Pooled NNK and combinatorial libraries were constructed using NNK primers (IDT) or oligo pools (Twist Biosciences), respectively. Combinatorial libraries were designed using one codon per amino acid (A = GCA, C = TGC, D = GAC, E = GAG, F = TTC, G = GGT, H = CAC, I = ATA, K = AAA, L = CTC, M = ATG, N = AAT, P = CCT, Q =

CAG, R = CGG, S = AGC, T = ACC, V = GTC, W = TGG, Y = TAT). In certain library compositions, cysteine (C) and methionine (M) were excluded due to their inherent reactivity and susceptibility to oxidation (for any given two position combinatorial library, $N = 324–400$). Variants and libraries were cloned into the digested empty vector at the BamHI/NsiI cut sites using standard Gibson assembly protocols. Sequences of an empty (MT) and a representative assembled (i53v_L67R) i53 lentiviral vector are provided in Supplementary Data 5.

## Cell culture
Lenti-X HEK293T cells (Takara Bio) were cultured in DMEM (1X) + GlutaMAX-I (Gibco) supplemented with 10% FBS (Sigma). K562 cells (ATCC) were cultured in RPMI (Gibco) supplemented with 10% FBS and 1x penicillin–streptomycin (Gibco). All cells were grown in a humidified 37 °C incubator with 5% $CO_2$ and were passaged every 3–5 d.

## Lentiviral production
Lenti-X HEK293T cells (Takara Bio) were seeded at a density of $4.5 \times 10^6$ cells per 10 cm dish 18–24 h prior to transfection. The prepared cells were co-transfected using the TransIT®-Lenti transfection reagent (Mirus Bio) with MISSION® Genomics Lentivirus Packaging Mix (Mirus Bio) and lentiviral plasmids containing i53 variants/libraries of interest. The viral supernatant was collected 48 h after transfection, passed through a 0.45 μm filter (Cytiva), flash frozen, and stored until use at -80 °C. Viral titers were measured by FACS in K562 cells and were typically ~0.5–1.5 × $10^7$ TU/mL.

## AAV production
The HBB-targeting AAV6 vectors HBB-SNP and HBB-UbC-GFP have been previously described[20,21,26,27]. All other AAV6 vectors were cloned into the pAAV-MCS plasmid (Agilent Technologies), which contains inverted terminal repeats (ITRs) derived from AAV2. Left and right homology arms (LHAs/RHAs) were derived from human genomic DNA to match the indicated length at the respective knock-in sites. The left and right homology arm lengths for the HBB, HBA, CCR5, and IL2RG donors were as follows: HBB LHA: 556 bp, HBB RHA: 449 bp, HBA LHA: 976 bp, HBA RHA: 879 bp, CCR5 LHA: 502 bp, CCR5 RHA: 500 bp, IL2RG LHA: 400 bp, IL2RG RHA: 414 bp. Each vector contained a UbC promoter, a CopGFP or (mCherry for HBB-UbC-mCherry), and a BGH polyA. UbC-GFP-BGH and UbC-mCherry-BGH were synthesized as gene fragments (Twist Bioscience) and cloned into pAAV with the corresponding LHA and RHA using standard Gibson Assembly protocols. The assembled LHA-UbC-GFP(/mCherry)-BGH-RHA sequences for HBB-UbC-mCherry, HBA-UbC-GFP, CCR5-UbC-GFP, and IL2RG-UbC-GFP AAV donors are provided are provided in Supplementary Data 5. The NPM1-GFP AAV6 vector was designed using the sequence of a donor plasmid described by the Allen Institute for Cell Science[45] which attaches an mEGFP tag to the C-terminus of NPM. LHA-linker-mEGFP-BGH-RHA was synthesized as a gene fragment (Azenta/Genewiz) and cloned into pAAV using standard Gibson Assembly protocols. The HBB-SNP AAV6 was produced by Viralgen. The HBA-UbC-GFP AAV6 was produced by Packgene. HBB-UbC-GFP AAV6, CCR5-UbC-GFP AAV6, IL2RG-UbC-GFP AAV6 and NPM1-GFP AAV6 were produced by Vigene. Titers used for CD34 + HSPC editing experiments were determined using droplet digital PCR (ddPCR).

## CD34+ HSPCs culture
Human CD34+ HSPCs were cultured as previously described[20,21]. CD34+ HSPCs were purchased from AllCells and had been isolated from G-CSF-mobilized peripheral blood from healthy donors. CD34+ HSPCs were cultured at $2.5 \times 10^5–5 \times 10^5$ cells/mL in StemSpan™-AOF (Stemcell) supplemented with stem cell factor (SCF) (100 ng/mL), thrombopoietin (TPO) (100 ng/mL), FLT3–ligand (100 ng/mL), IL-6 (100 ng/mL) (all Peprotech) and UM171 (35 nM) (Selleckchem). Cells were cultured at 37 °C, 5% $CO_2$, and 5% $O_2$.

## Lentiviral transduction of CD34+ HSPCs
CD34 + HSPC cells were transduced using lentivirus at MOIs of 0.25–1 at day 1 post thaw. Cells were concentrated using centrifugation (180 x g, 7 min), counted, and added at a concentration of $4 \times 10^6$ cells/mL to media containing lentivirus, cyclosporin A (5uM, Sigma Aldrich), and Synperonic F108 (0.5 mg/mL, Sigma Aldrich). After 4 h of incubation, cells were spun down, washed once with media, and seeded into lentivirus-free media at a density of $3.5 \times 10^5$ cells/mL.

## Genome editing of CD34+ HSPCs (AAV6 donor)
Chemically-modified single guide RNAs (sgRNAs) used to edit CD34+ HSPCs were purchased from Synthego. The sgRNA sequences were modified by adding 2′-O-methyl-3′-phosphorothioate at the three terminal nucleotides of the 5′ and 3′ ends. The target sequence for the sgRNAs used are as follows: *HBB*: 5′-CTTGCCCCACAGGGCAGTAA-3′, *HBA*: 5′-GGCAAGAAGCATGGCCACCG-3′, *CCR5*: 5′-GCAGCATAGTGAG CCCAGAA-3′, *IL2RG*: 5′- TGGTAATGATGGCTTCAACA-3′, and *NPM1*: 5′-TCCAGGCTATTCAAGATCTC-3′. Cas9 protein (SpyFi Cas9) was purchased from Aldevron. The RNPs were complexed at a Cas9: sgRNA molar ratio of 1:2.5 at 25 °C for 10–15 min prior to electroporation. 48–72 h post thaw, CD34 + HSPC cells were collected, counted, and pelleted at 180 g x 7 min. The cell pellets were resuspended in MaxCyte buffer (standard cell concentrations per cuvette, as recommended by vendor, are shown in Table M1 in Supplementary Data 6) with complexed RNPs (final concentrations in electroporation cuvette: 0.45 mg/mL Cas9, 0.24 ug/uL sgRNA) and electroporated using a MaxCyte ExPERT ATx Nucleofector. After electroporation cells were plated at $3.5–5.0 \times 10^5$ cells/mL in media supplemented with cytokines and the desired AAV6 donor added at $5.0 \times 10^2–2.5 \times 10^4$ vector genomes/cell. 24 h after nucleofection, cells were spun down, washed once with media, and seeded into AAV-free media at a density of $3.5 \times 10^5$ cells/mL. Cells were harvested 1–2 days post nucleofection for NGS analysis (see figure captions for specific times) or 3–5 days post nucleofection for GFP expression analysis.

When editing using purified i53 variant proteins, the proteins were added to CD34+ HSPCs cells as part of the nucleofection mix at concentrations of 0.0125–1.6 mg/mL (volume of added protein ≤ 1/10 of MaxCyte cuvette volume) prior to nucleofection. For editing with a DNAPK small molecule inhibitor (AZD7648, CC-115, and M314/nedisertib from Selleck Chemicals, or BAY8400 from MedChem Express), nucleofected cells were added to media containing both AAV6 and the DNAPKi at various concentrations. Twenty four hours after nucleofection, cells were spun down, washed with media, and seeded into AAV6 and DNAPKi-free media at a density of $3.5–5.0 \times 10^5$ cells/mL.

## Screening and sorting of pooled libraries
Lentiviral-based i53 variant libraries were transduced at an MOI of ~0.2–0.5 (aiming for ~ 30% transduction and a coverage of >500 cells per library member in mCherry +/GFP+ cell population for each replicate tested). Three days after transduction, cells were edited in triplicate or quadruplet at *HBB* (or *NPM1*) as described above, using HBB-UbC-GFP donor AAV6 (or NPM1-GFP donor AAV6) at an MOI of $2.5 \times 10^4$ vector genomes/cell. Three days post editing, cells were pelleted and resuspended in media with DAPI (Miltenyi Biotec). Single, live, mCherry +/GFP+ and mCherry +/GFP- cells were collected using a FACSAria cell sorter (Becton Dickinson); purity of populations was confirmed by post-sort purity checks. Post sort, genomic DNA was harvested from each sorted cell population using a Quick-DNA 96 Plus Kit (Zymo Research). The DNA concentration of each sample was measured using a Qubit 1X dsDNA BR assay kit (ThermoFisher).

## Next Generation Sequencing (NGS) of pooled libraries
An amplicon sequencing workflow was designed to sequence and quantify i53 variants within starting and post selection pools. Primers and PCR conditions were optimized to specifically amplify the entire

variant coding sequence from plasmids, lentiviral libraries, as well as genomic DNA carrying lentiviral vector insertions. After the initial amplification, the i53 amplicons undergo an additional PCR amplification to add sequencing adapters and sample indexes to enable sample multiplexing. The resulting sequencing libraries were then sequenced on an Illumina MiSeq instrument using paired end reads to cover the full length of the i53 coding sequence.

## NGS analysis of pooled libraries

The frequencies of i53 variants were quantified by counting the number of each observed sequence in the NGS data and then removing all unexpected sequence (i.e. using a prespecified "whitelist" of variants known to be contained in the pool). Spike-in tests using individual variants demonstrated the sequencing and analysis workflows could correctly estimate the frequencies of different i53 versions. This approach was used to confirm sequence diversity in plasmid and lentiviral libraries prior to screening. For quality control of screening data, key measures we considered were: the number of mapped reads (>1e5 reads per sample), the percent reads carried over from parent, and the diversity of observed sequences (i.e. minimal skewing). Fold-change enrichment of a variant was calculated by dividing normalized variant frequency in GFP+ cells by the frequencies in GFP- cells sorted from the same parent. All datasets contained an internal control (NNK-generated parent sequence) that was used to perform a last quality control of datasets, excluding sequencing runs where internal control abundance was >10% different from that of parent carry over control.

Data processing and visualizations were generated using R (v4.1.2) and the *ggplot2* package. Variants were ranked by fold change over parent and any variant for which either average or every replicate was over 1.0 was flagged as 'Better than parent', as highlighted in figures. Hits were ranked by average fold change and top selected candidate variants were moved forward for validation in targeted libraries, as described below.

## Validation of hits via lentiviral expression

Sequences of individual i53 variants of interest were cloned into the lentiviral-based expression plasmid described above. Hits were validated either as pooled "validation libraries" (variant and control plasmids manually mixed to generate a pool of 5−25 variants) or individually. Lentivirus generated from these plasmids was used to transduce CD34 + HSPC cells at MOIs of 0.5−1 at day 1 post thaw. At day 4, the transduced cells were edited with HBB-UbC-GFP AAV6 at concentrations of 1.25−2.5 × 10$^4$ vector genomes/cell. Cells transduced with pooled validation libraries were edited in triplicate or quadruplet; cells transduced with individual variants were edited in duplicate.

For individual testing of variants, rates of integration of the HBB-UbC-GFP donor were measured using a Beckman Coulter CytoFLEX. DAPI (Miltenyi Biotec) was used to discriminate live and dead cells. mCherry expression was used to differentiate transduced cells from untransduced cells and rates of GFP integration were compared between the two populations to quantify the impact of lentiviral-based variant expression on HDR rates. Flow cytometry data were analyzed using FlowJo 10 software. For pooled validation libraries, cells were sorted and analyzed as described above. NGS analysis of the gDNA purified from sorted mCherry+GFP+ and mCherry+GFP- populations was used to determine differential variant enrichment and validate the impact of individual variants on HDR rates relative to a control.

## i53 variant protein production and purification

The sequences of different i53 variants were cloned into bacterial expression plasmids, resulting in a N-terminal His-tagged fusion protein with a protease cleavage site in between the 6x-His-tag and i53 variant sequence. The resulting plasmids were transformed into *E. coli* BL21 (DE3)-RIL for protein expression. Cells were grown at 37 °C in Luria-Bertani broth supplemented with 0.4% glucose to OD600 = 0.8

and induced with 0.4 mM IPTG at 18 °C for 18 h. Cells were harvested by centrifugation, resuspended in 50 mM potassium phosphate pH 8.0, 500 mM NaCl, 20 mM imidazole, and 3 mM β-mercaptoethanol. Cells were lysed using an microfluidizer (Microfluidics). The crude lysate was immediately supplemented with 0.2 mM phenylmethylsulfonyl fluoride (PMSF) and centrifuged at 14,000 x g for 30 min. The soluble fraction was subsequently incubated with 2 ml Ni-NTA (GE Healthcare) per 1000 ODs for 1 h at 4 °C. Following incubation with the Ni-NTA resin, lysate was removed by pelleting the resin at 2500 g for 3 min and washed 3 times with 9 bed volumes of 50 mM potassium phosphate pH 8.0, 500 mM NaCl, 20 mM imidazole, and 3 mM β-mercaptoethanol. Following the batch wash Ni-NTA resin was loaded onto a gravity column and His-tagged i53 variant protein was eluted with 6 bed volumes of 50 mM potassium phosphate pH 8.0, 300 mM NaCl, 500 mM imidazole, and 3 mM β-mercaptoethanol. Eluted protein was dialyzed overnight against 10 mM Tris/HCl pH 8.0, 200 mM NaCl, and 1 mM DTT and the 6xHis-tag was cleaved with protease. The protein was purified by anion exchange chromatography on a HiTrapQ column (GE Healthcare) via a linear NaCl gradient and twice by size exclusion chromatography using a Superdex S200 26/60 column (GE Healthcare) run in 10 mM Tris/HCl pH 8.0, 200 mM NaCl, 1 mM DTT. Proteins were concentrated to ~20 mg/mL and flash frozen for storage.

## Size exclusion chromatography

Recombinantly purified 53BP1 Tudor domain (53BP1 residues 1484−1603) was mixed with recombinantly purified i53 variants at a concentration of 0.5 mg/mL each. Proteins were incubated for 30 min at room temperature prior to injection onto an HPLC (Agilent, 1260 Infinity II). 5 µL of protein complex was injected onto a MAbPac 4 × 300 mm SEC column with 5 µm particle size and 300 Å pore size. The HPLC was run at 0.2 mL per minute using PBS as the mobile phase and continuously measuring the absorbance at 280 nm for ~1 full column volume. 53BP1 Tudor domain alone has a retention time of 14.6 min. i53 variants have a retention time of ~15.5 min. A stable complex of 53BP1 Tudor domain and i53 variants were found to have a retention time of 14.3 min.

## Bio-layer interferometry (BLI)

Data were collected using an Octet R8 system (Sartorius). Purified 53BP1 Tudor domain was labeled at exposed primary amine groups with NHS-biotin using ChromaLINK NHS-Biotin protein labeling kit (Vector Laboratories). 1 equivalent of chromalink biotin was incubated with the 53BP1 Tudor domain for 2 h and buffer exchanged into fresh PBS. Labeling efficiency was calculated to be ~1 biotin per molecule of 53BP1 Tudor domain. Octet SA Biosensor tips (Sartorius) were incubated with biotin-labeled 53BP1 Tudor domain (ligand) for 60−80 s. The labeled tip was then dipped in 1x binding buffer (Sartorius) for 60 s to remove excess ligand and achieve baseline. Labeled tips were introduced to the i53 variant (analyte) for 500−600 s and the response was continuously monitored to detect association. A range of analyte concentrations were tested from a highest to lowest concentration in nM (i.e. 200, 100, 50, 25, 12.5, 6.25, 3.125). The tips were then introduced to 1x binding buffer for 5 min and the response was continuously monitored to detect dissociation. A dissociation constant ($K_D$) was calculated using a 1:1 binding model and the on-rate ($k_a$) and off-rate ($k_d$) were calculated as a change in response (nm) over time (s).

## TR-FRET

Assay volumes of 20 µL ($n = 4$) were composed of 0.5 uM His-tagged i53, 0.5 µM c-terminal avi tagged 53BP1, 5 nM Europium labeled anti-His antibody (Perkin Elmer), 0.5x Streptavidin-xl665 (Cis Bio) and an i53 variant at concentrations ranging from 5000 nM to 4.8 nM. All assay components were prepared in a buffer composed of 50 mM Tris pH 7.5, 150 mM NaCl, 0.02% (v/v) Tween-20, and 0.05% (w/v) BSA. Each

assay was incubated for 2 h at room temperature in a 384 well white optiplate. TR-FRET was measured on a Clariostar Plus plate reader (BMG LabTec) using the TR-FRET mode.

## Crystallography

The human i53:53BP1 complex, purified in 10 mM Tris 8.0, 200 mM NaCl and 1 mM DTT was screened for crystallization at room temperature using a protein concentration of 30 mg/mL with the previously published condition[15] 0.1 M MES (2-(N-morpholino) ethanesulfonic acid) pH 6.0, 0.2 M trimethylamine N-oxide and 25% (w/v) PEG MME (polyethylene glycol monomethyl ether) 2000. Crystals grew within 7 days at 23 °C using the sitting drop vapor diffusion method. Crystals were cryoprotected by adding glycerol, 20% (v/v) final concentration, to the reservoir solution before flash-freezing in liquid nitrogen. The i53:53BP1 complex was crystallized in the $P2_12_12_1$ space group with one i53:53BP1 complex molecule per asymmetric unit cell.

## Structure determination

X-ray diffraction data was collected at the CLSI beamline 081D-1 using a wavelength of 0.95372 Å under cryo-conditions at a temperature of 100 K. Structures of human i53:53BP1 Tudor domain (WT, L67H, L67R, T12Y.T14E.L67R, T12V.T14H.L67H) were solved using molecular replacement and previously published structure of WT i53:53BP1 Tudor domain (PDB code: 5J26). The final models for human i53:53BP1 Tudor domain (WT, L67H, L67R, T12Y.T14E.L67R, T12V.T14H.L67H) were built with native data and refined to an extended resolution below 1.8 Å for each dataset. All models of i53:53BP1 complex were built using COOT[46] and further refinement was completed using Refmac[47]. All models were refined to acceptable quality and Ramachandran values are reported below for each dataset. Ramachandran values were calculated for each dataset and are as follows. i53WT:53BP1—98.94% favored, 1.06% allowed, and zero outliers. i53L67R:53BP1—98.86% favored, 1.04% allowed and zero outliers. I53L67H:53BP1 98.45% favored, 1.55 % allowed and zero outliers. i53T12V.T14H.L67H:53BP1—97.4% favored, 2.6% allowed, and zero outliers. i53T12Y.T14E.L67R:53BP1—97.28% favored, 2.72% allowed and zero outliers. The respective PDB codes are 8SVG, 8SVH, 8SVI, 8SVJ, 8T2D.

## LC-MS

Samples of purified proteins (20 μg) were analyzed by LC-MS using a Poroshell 300SB-C8 2.1 × 7.5 mm column coupled to an Agilent 6224 ToF (data collection and analysis completed at JadeBio, San Diego, CA).

## Measuring targeted integration of HBB-UbC-GFP, HBA-UbC-GFP, CCR5-UbC-GFP, or IL2RG-UbC-GFP (flow cytometry-based analysis)

Rates of targeted integration of the HBB-UbC-GFP, HBA-UbC-GFP, CCR5-UbC-GFP, and IL2RG-UbC-GFP donors were measured using a Beckman Coulter CytoFLEX. DAPI (Miltenyi Biotec) was used to discriminate live and dead cells. Flow cytometry data were analyzed using FlowJo 10 software.

## Measuring targeted integration of HBB-SNP (NGS-based analysis)

The frequency of homology directed repair (HDR) and other editing outcomes at *HBB* were measured using Next Generation Sequencing (NGS). An NGS assay was developed to determine the frequency of various sequence changes at the *HBB* locus by quantifying the number of alleles that have been either: (1) not edited (% WT), (2) changed by HDR to incorporate sequence differences present in the AAV repair template (% HR), or (3) mutated during the genome correction process resulting in a gene that produces mutant β-globin (% INDELs).

For this assay, genomic DNA was harvested from cells using a Quick-DNA 96 Plus Kit (Zymo Research). The DNA concentration was measured using a Qubit 1X dsDNA BR assay kit (ThermoFisher). Purified genomic DNA was then used to amplify the *HBB* locus via polymerase chain reaction (PCR). The PCR products were diluted using nuclease-free water to serve as the template DNA for targeted NGS library prep. An Aglient Tapestation was used to confirm the PCR product for each sample was the expected size (1410 bp). A second PCR with primers carrying partial Illumina adapters was performed to amplify a 142 base pair sequence that includes the region of the *HBB* locus that is to be corrected during the genome correction process. The PCR products were diluted again to serve as templates in a third PCR reaction using Nextera XT index primers. This third PCR reaction was used to assign unique identifiers to each sample and to add the full length adapter sequences necessary for Illumina sequencing. The size of the PCR products was assessed on an Agilent BioAnalyzer. PCR products were then pooled, purified using a Qiagen PCR purification kit, and quantified using PicoGreen in order to ready the PCR products for sequencing.

Based on the PicoGreen concentration, the library of pooled PCR products was diluted to a final concentration of 4 nM. Sequencing was performed on a MiSeq system using an Illumina MiSeq sequencing reagent kit (V2, 300 cycles). A 10% PhiX control library was added to the sample library to improve sequence diversity and to allow for error rate measurements. The library was denatured and loaded at 8–12 pM onto the sequencing reagent cartridge. The sequencing entails paired-end 150 base pair reads and dual indexing reads. The sequencing data was demultiplexed based on the sample indexes provided and FASTQ files for each sample were generated. The FASTQ files were processed using the CRISPResso2 pipeline (v2.1.0)[48] or all experimental and bioinformatic steps, a positive control with known editing outcomes, a negative control with no editing and a no template control were processed in parallel with each set of samples.

As has been reported previously, recombination events were observed where double-stranded breaks at the *HBB* locus were repaired with *HBD*, a close and nearby homolog of *HBB*. These various recombination events could be recognized by the presence of up to 6 SNPs only present in *HBD* and not the HBB-SNP repair template nor the *HBB* wildtype sequence. To estimate the frequency of *HBB* break repair using *HBD* as a template the fully recombined HBD amplicon sequence was included (containing all 6 mismatch SNPs relative to the *HBB* amplicon) as an amplicon in Crispresso (in addition to wildtype *HBB* and the intended repair outcome with HBB-SNP using the "-a" parameter).

To quantify partial recombination (I.e. containing <6 mismatch SNPs), during Crispresso analysis we generated two amplicon sequences consisting of 5 of the 6 HBD-specific SNPs in the 3' direction and 5 of the 6 SNPs in the 5' direction from the cutsite. To quantify *HBD* recombinations, we summed the number of reads that mapped to either the full or the partial *HBD* recombination plus reads containing a mismatch at the cutsite without any additional indel (all of those tracked to *HBD* gene).

The full list of parameters passed to Crispresso was the same for all analyses and are shown in Table M2 in Supplementary Data 6.

Summary data for each sample was reported as % WT (unedited), % HDR (incorporation of HBB-SNP donor template), % HBD, % MMEJ (edits that get significantly reduced by POLQ knockdown, as described below), while the rest are classified as "NHEJ". When presented as % edited alleles, edits are calculated as % of any given edit/(100−WT).

## Measuring editing of OT-1 (NGS-based analysis)

Editing outcomes at the off-target (OT) editing site OT-1 were assessed using an assay very similar to the one described above for measuring the targeted integration of HBB-SNP. The off-target editing site was identified via three different methods (in silico prediction[26], Circle-Seq[49], and Guide-Seq[50]) and was confirmed to be off-target editing site

of significance via amplicon sequencing. The workflow for the OT-1 amplicon sequencing assay is very similar to the on-target sequencing assay but involves one less PCR step. A small 166 base pair sequence encompassing the OT-1 editing site is amplified directly from genomic DNA in the first PCR and then tagged with barcode and adapter sequences in a second PCR. The resulting PCR products are sized, quantified, and prepared for sequencing in the same manner as for the *HBB* on-target sequencing assay. The sequencing data is also processed using the Crispresso2 pipeline[47], but in contrast to the on-target sequencing assay no homology direct repair outcomes are present at OT-1, only % WT and % INDELs are reported.

### DNA damage response (DDR markers p21 and yH2AX) analysis

For p21 analysis, $1.5 \times 10^5$ cells were spun down at 300 g x 5 min, washed once with PBS, and was resuspended in 22.5 µL of RIPA buffer with 2X Halt protease and phosphatase inhibitors (ThermoFisher). Lysates were incubated on ice for 30 min with intermittent vortexing. Lysates were then spun down in a microcentrifuge at 500 x g for 5 min; supernatants were then transferred to fresh tubes. Samples were prepared by mixing 5 µL of protein extract with 1.25 µL of freshly prepared 5X fluorescent master mix as instructed by the ProteinSimple Jess protocol. Samples were denatured for 10 min at 95 °C, quickly spun and loaded into a Jess capillary cartridge. Capillaries were probed with anti-p21 (CST) and anti-alpha tubulin (Abcam) and detected by HRP-conjugated secondary antibodies. Data was normalized to the internal alpha-tubulin loading control and then expressed as FC values over control treatments.

For yH2AX analysis, $1.5 \times 10^5$ cells were spun down at 300 g x 5 min. Cells were resuspended in 100 µL of diluted Live/Dead Fixable Violet dye (ThermoFisher, 1:1000 in PBS). Cells were incubated for 20 min in the dark at room temperature. After adding 100 µL of PBS to cell suspension, the cells were spun down and washed once more with 200 µL PBS. The final PBS wash was flicked from the plate and cells were lightly vortexed to resuspend in residual PBS buffer remaining after wash. 100 µL of freshly-prepared 70% ethanol was added to the cell pellets and the plate was tightly sealed with foil, vortexed and then allowed to fix at −20 °C from 1 h to 3 days. Following fixation, 100 µL of cell staining buffer (CSB, BioLegend) was added to cell suspensions and the resuspended cells were spun down at 500 x g for 5 min. Cells were washed once more with 200 µL of CSB and then resuspended in 50 µL of CSB and blocked for 15 min at RT. Diluted anti-yH2AX-PE (1:20 in CSB, BioLegend) was added to cells and incubated further for 30 min at RT. Following staining, cells were washed twice with CSB and were immediately analyzed using a Beckman Coulter CytoFLEX flow cytometer.

### Apoptosis assay

For each sample, $1.2 \times 10^5$ total cells in growth medium were centrifuged for 5 min at 400 x g, the supernatant was aspirated, and cells were resuspended in 100 µL of 1x Annexin V Binding Buffer (ABB, Fisher Scientific) supplemented with 5 µL of Annexin V-AF488 reagent (Fisher Scientific). After 15 min incubation at RT hidden from light, 3 µL of 1:10 dilution of 1 mM Sytox AAD stock in DMSO (Fisher Scientific) was added and mixed well. After 5 min incubation, 150 µL of 1x ABB was added and cells were immediately analyzed on CytoFLEX LX (Beckman Coulter) using B525-FITC and R712-APCA700 channels. For negative and positive staining controls, untreated cells or cells cultured for 20–24 h in complete medium supplemented with Etoposide (R&D Systems; 5 µM for 16 h) were stained in parallel with samples, respectively. The flow cytometry results were analyzed using FlowJo v10.8 Software (BD Life Sciences). The compensation matrix was built in FlowJo using single-stained control cells. Before quantation of viable and apoptotic cell populations, cell debris was gated out from the double negative population.

### POLQ knockdown and determination of MMEJ edits

The construct for the shRNA knockdown of PolQ was adapted from the previously reported pLKO[51] and cloned from a third-generation lentiviral plasmid (Lenti SFFV) purchased from Twist Biosciences. An empty vector was constructed to include a DNA stuffer flanked by AgeI and EcoRI restriction enzyme cut sites downstream of a U6 promoter and upstream of a SFFV-EGFP-WPRE cassette (to enable fluorescence-based monitoring of cells expressing the shRNA). shRNA sequences were cloned as duplexed DNA oligos (IDT) into the digested empty vector at the AgeI/EcoRI cut sites using standard Ligation protocols. The target sequences used for *POLQ* (gene ID: 10721) and non-targeting control (NTC) shRNA were identified using the Broad Genetic Perturbation Platform (GPP) Web Portal (https://portals.broadinstitute.org/gpp/public/gene/search) and are as follows were: POLQ: 5′-GCTGACCAAGATTTGCTATAT-3′ and NTC: 5′-CCTAAGGT-TAAGTCGCCCTCG-3′. Sequences of an empty (MT) and a representative assembled (Polq) shRNA lentiviral vector are provided in Supplementary Data 5.

CD34 + HSPC cells were transduced using lentivirus produced using shRNA transfer vectors at day 1 post thaw using methods described above and MOIs of 2.5–7.5. Three days after transduction, cells were edited in duplicate at *HBB* as described above, using HBB-SNP or HBB-UbC-mCherry donor AAV6 (MOIs of 75–2.5 × 10^4 vector genomes/cell). Three days post editing, cells were pelleted and resuspended in media with DAPI (Miltenyi Biotec).

For cells edited with HBB-UbC-mCherry: rates of integration of the donor were measured using a Beckman Coulter CytoFLEX. DAPI (Miltenyi Biotec) was used to discriminate live and dead cells. GFP expression was used to differentiate transduced cells from untransduced cells and rates of mCherry integration were compared between the two populations to quantify the impact of lentiviral-based shRNA expression on HDR rates.

For cells edited with HBB-SNP: single, live, GFP+ (shRNA + ) and GFP- (shRNA-, negative control) cells were collected using a FACSAria cell sorter (Becton Dickinson); purity of populations was confirmed by post-sort purity checks. Post sort, genomic DNA was harvested from each sorted cell population and editing outcomes were determined using the pipeline outlined above. To determine which edits were reduced by POLQ knockdown, a one-sided *t*-test comparing the GFP+ and GFP- conditions for each individual editing outcome. Those edits with FDR-corrected (Benjamini-Hochberg) *p*-value below 0.1 were labeled as POLQ-dependent on "MMEJ".

### LT-HSC sorting Method and Materials

Details on the antibodies and reagents used for LT-HSC sort are shown in Table M3 in Supplementary Data 6. Cryopreserved samples were rapidly thawed in warm GMP SCGM (CellGenix) media, washed with cell staining buffer (Biolegend). Washed cells were incubated with a panel of fluorochrome-conjugated anti-human monoclonal antibodies (mAb) and viability dye to characterize hematopoietic stem cell compartments. The following directly conjugated mAbs used in this study were obtained from BD Biosciences: CD38-PE-Cy7 (Clone HIT2), Biolegend: CD34-Aexa488 (581), CD45RA-BV510 (HI100), CD49c-PE (ASC-1), CD49f-BV421 (GoH3), CD90-BV711 (5E10), CD201-APC (RCR-401), CD45-Alexa700 (HI30) and Thermo Fisher Scientific: DyLight 800 Maleimide. Brilliant Stain Buffer Plus (BD Bioscience) was added to stabilize the fluorophore-conjugated antibody cocktail. Cells were stained for 30 min at 4 °C and washed with cell staining buffer and acquired within 1 h on a custom SORP five laser FACSAria Fusion (BD Biosciences).

FACSAria Fusion was calibrated with Cytometer Setup and Tracking beads (BD Biosciences, 655050), the sort parameters were set to 20 psi with a 100 µm nozzle, and the droplet stream was calibrated with Accudrop Beads (BD Biosciences, 345249). Sort layout was set to

4-way purity and four subfractions were collected into 5 mL FACS tubes as follows:

1. Long Term Hematopoietic Stem Cell (LT-HSC) enriched (CD34 + CD45RA-CD90 + CD201 + CD49f + CD49c + )
2. Short Term hematopoietic Stem Cell (ST-HSC) enriched (CD34 + CD45RA-CD90 + CD201- (CD49f-CD49cdim))
3. Hematopoietic Stem and progenitor cell (HSPC) enriched (CD34 + CD45RA-CD90-CD201- (CD49f-CD49c-))
4. Lineage committed progenitors (CD34 + CD45RA+ (CD90dimCD201-CD49f-CD49cdim))

Aliquots of sorted sample populations were re-acquired to assess the purity of the sort. Sorted cells were spun down, supernatant aspirated, and snap frozen at -80 °C for DNA extraction and NGS analysis.

The bulk cells were phenotyped for cell sorting with the surface markers CD45, CD34, CD45RA, CD201, CD90, CD49f, CD49c[9,38-40]. A physical gate was applied to remove debris and isolate HSC sized cells, doublet cells were removed with SSC-singlet and FSC-singlet gates, dead cells were removed, the CD45+ cells were subfractionated into HSC/HSPC (CD45 + CD34 + CD45RA-) and linage committed (sort population 4: CD45 + CD34 + CD45RA + ) compartments. The HSC/HSPC was further divided into HSPC (sort population 3: CD45 + CD34 + CD45RA-CD90-CD201-(CD49f-CD49c-)), short-term HSC (sort population 2: CD45 + CD34 + CD45RA-CD90 + CD201- (CD49f-CD49cdim)) and long-term HSC (sort population 1: CD45 + CD34 + CD45RA-CD90 + CD201 + CD49f + CD49c + ).

## CFU progenitor assay

At 48 h post gene editing, 250 cells per well were plated in Methocult Optimum media in SmartDish plates (both StemCell Technologies). Plates were incubated in a secondary enclosure at 37 °C, 5% $CO_2$, and 5% $O_2$ for 14 d before scoring colonies using the human mPB program on a STEMvision imager (StemCell Technologies).

## Measuring targeted integration of HBB-SNP in Colonies (CFU-seq)

Individual colonies were picked and gDNA was extracted using Lucigen Quickextract kit according to manufacturer's instructions. NGS library prep on gDNA was performed as described in above section titled Measuring Targeted Integration of HBB-SNP.

Raw fastq files output from the sequencer were analyzed using our in-house On-target HBB CFU Bioinformatics Pipeline. This pipeline uses Crispresso 2 (v2.1.0) to quantify the various gene editing outcomes in each colony. Parameters for Crispresso 2 were set to be identical to those used for On-Target CD34 NGS analysis. Output counts and fractions of each allele from Crispresso analysis was used to infer genotypes. Filters were applied to remove low quality colonies. Colonies with fewer than 2000 reads aligned were removed as the low read count would likely impact quantification. A 10% fraction threshold was used to call the presence of expected alleles. Colonies with more than two alleles above the 10% fraction threshold were removed as these were likely not single clones. NoCall colonies included any colonies that did not produce a band on In/Out PCR or were removed by the above bioinformatics filters.

## Guide-seq

Genomic DNA (gDNA) was isolated using PureBind Blood Genomic DNA Isolation Kit (Ocean Nanotech), quantified, library preparation was performed, and quality was assessed (LAB-SOP-018, GeneGoCell) then sequenced (NextSeq2000, Illumina). Raw sequence reads were demultiplexed into sample-specific fastq files (bcl2fastq program v2.20.0.422, Illumina). The resulting fastq files were processed as follows: low-quality reads were removed using quality score threshold 28 (Q28), and PCR duplicates were removed using the UMIs. The resulting fastq files were analyzed to generate quality control (QC) statistics.

Reads were aligned to the human genome (hg38) using BWA v0.7.17-r1188 (GeneGoCell NGS bioinformatics pipeline v2.2.3).

The control and experimental samples were further analyzed using the same process, abbreviated here: For a given site, the dsODN insertion rate was calculated as the number of site-specific reads with dsODN incorporation vs. total number of site-specific reads. The alignment results were analyzed using G-GUIDE analysis program v4.0 to generate the genome-wide dsODN insertion sites and report break points (BPs) for each high-quality read. Control sample background sites were subtracted from the edited samples, and only sample-specific sites are reported.

## Karyotyping

Cryopreserved aliquots of 2 M cells were used for each submission. Aliquots of 2 M cells for each sample (24 h post-editing) were prepared by centrifuging cells (180 x g, 7 min) and resuspending in Cryostor CS10 solution (BioLife Solutions) at a density of 10 M cells/mL. Frozen cell aliquots were then sent to KromaTiD (Longmont, CO) for karyotyping (G-banding). Briefly, after harvest and fixation, the fixed cells were washed twice with fixative (prepared fresh day-of-use) and the O.D. was adjusted. Drops of the final cell suspension were placed on clean slides and aged for 60 min at 90 °C. Slides were digested in a pancreatin solution with Isoton II diluent. The enzymatic reaction was then stopped by rinsing with FBS, followed by application of a stain solution (3:1 Wright/Gurr buffer) which was poured on the slides so that it covered the entire surface. After staining for up to 1 min, slides were washed with de-ionized water for 1−5 s and air dried. A mounting medium was applied to the slides and sealed with a coverslip. The slides were scanned on the microscope for cell analysis.

## Translocation-seq

Detection of the sequence of interest and their translocated partners, in this case our editing site, known off-target site and their translocation partners. Samples were treated as follows: gDNA was isolated (PureBind Genomic DNA Isolation Kit, Ocean Nanotech), fragmented via sonication, followed by DNA-end repair, UMI adapter ligation, and PCR amplification enrichment of fragments that contain editing targets and translocations, then prepared for sequencing (LAB-SOP-017, GeneGoCell). The amplified gDNA fragment library was sequenced (NextSeq 2000, Illumina) and DNA sequence generation via sequencing-by-synthesis (SBS) (LAB-SOP-022, GeneGoCell), demultiplexed (bcl2fastq v2.20.0.422, Illumina) and processed as described next (v2.0.9, GeneGoCell):

GeneGoCell's proprietary G-Trans platform was used to amplify and quantify all potential translocations in an unbiased manner. Translocations were quantified between the target listed below to anywhere else across the genome (hg38). Low quality reads and PCR duplicates were removed via Q28 and UMIs, respectively. Quality control was run (v.0.10.1, FastQC), reads were aligned to hg38 (v0.7.17-r1188, BWA), and results were analyzed (proprietary translocation analysis v1.6, GeneGoCell) to identify potential genome-wide translocation sites. Donor and recipient genomic loci BPs were calculated per read. Called BPs met the following CRISPR/Cas9 genome editing associated criteria: ≥10 UMI reads, a minimum of 3 BPs in the flanking 200 bp of the peak, position +/- 100 base pairs, and peak BP:total region read counts ratio <0.9. To compute on-target translocation rate, the number of reads for each reaction was divided by the number of target-specific reads, and multiplied by 100.

Details on the targets and primers used for Trans-seq are shown in Tables M4 and M5 in Supplementary Data 6.

## scRNA-seq

To assess gene expression profiles from single cells, 2 million cryopreserved cells were thawed for use with 10X Genomics Chromium

Next GEM Single Cell 3' Gene Expression Reagent Kit (10 × 3' Kit). The thawed cells were counted with AO/PI viability stain on the Nexcelom cell counter. Approximately, 8000 live cells were added to a master mix for reverse transcription (RT Reagent B, Template Switching Oligo, Reducing Agent B, and RT enzyme C), then loaded into a Chromium Next GEM Chip G for running in the Chromium Controller to generate Gel Beads-in-emulsion (GEMs). The GEMs were transfer to tubes for RT incubation in a Bio-Rad C1000 Touch for 45 min at 53 °C, then 5 min at 85 °C and held at 4 °C. After RT, the GEMs were purified with Dynabeads™ MyOne™ SILANE. The eluted cDNA was amplified by using the Amp Mix and cDNA primers with 11 cycles of PCR in a Bio-Rad C1000 Touch. The dsDNA cDNA product was analyzed using the High Sensitivity DNA Chip on an Agilent Bioanalyzer 2100. 10 µL of the dsDNA cDNA product was fragmented with the fragmentation primer, end-repaired and A-tailed to prepare for the ligation of the sequencing adapters. Afterwards, the dsDNA was purified with a double-sided SPRI. Illumina sequencing adapters were ligated to the dsDNA to generate the sequencing library. Another 15 cycles of PCR in a Bio-Rad C1000 Touch was used to amplify and index the sequencing library.

The indexed libraries were purified with a double-sided SPRI and qualitatively measured with the High Sensitivity DNA Chip on an Agilent Bioanalyzer 2100 to assess the size and the concentrations were measured by Qubit Broad Range kit. Each sample library was normalized to 900 pM and pooled in equal volumes. The library pool plus including 10% PhiX control library was denatured and then loaded onto a P3 flowcell on the Illumina NextSeq 2000. The run parameters were Read 1: 28, Index 1: 10, Index 2: 10, Read 2: 90, per 10 × 3' Kit protocol. After the run, the sequence metrics was checked to see read quality and then bcl files were converted to fastq. The fastq were then input into the Graphite single cell pipeline for analysis.

Data processing and analysis were performed in R version 4.2.0 via RStudio, using Seurat (v4.3.0). Visualizations were created with dittoSeq (v1.8.1) (https://github.com/dtm2451/dittoSeq/) and ggplot2. Seurat's *Read10X* function was used to generate a count data matrix using the filtered count matrix genes and cells, gene names, and barcode files provided by 10X. A Seurat object was created with the count data matrix and metadata and filtered to keep genes present in at least 3 cells and cells meeting cohort selection criteria of at least 200 genes. Log normalization was performed using Seurat's *NormalizeData* function with a scale factor of 10,000, and highly variable features were identified using Seurat's *FindVariableFeatures* The data matrix was then scaled using Seurat's *ScaleData* function with nCount_RNA regressed out, and dimensionality reduction through Uniform Manifold Approximation and Projection (UMAP) was performed with the appropriate dimensions selected based on the corresponding principal component analysis (PCA) elbow plot.

The Seurat function *RunAzimuth* was used as reference-based mapping to annotate the data to the Human bone marrow reference (https://azimuth.hubmapconsortium.org/).

## Isolation of CD34+ cells
Leuokopaks were purchased from AllCells LLC and these were collected from healthy donors per standard protocols using mobilization with G-CSF+ Plerixafor. CD34+ cells were isolated from the leukopaks within 24 h by first removing the platelets using the LOVO cell Processing System (Fresenius Kabi). GMP grade reagents, buffers and columns for CD34 immunomagnetic selection were purchased from Milteny Biotec and the platelet washed cells were incubated for 30–35 min using the CD34 Reagent following which a subsequent wash for excess antibodies was performed on the LOVO. The washed and labelled cells were subject to immunomagnetic selection using the CliniMACS Plus instrument (Miltenyi Biotec) following which the cells were cryopreserved at a concentration of $5 \times 10^6$ - $1 \times 10^7$ cells/mL in CryoMACS 50 or 250 Bags (Miltenyi Biotec) for the gene edited drug product generation step.

## Large scale editing of HSPCs
At least $5 \times 10^7$ - $3 \times 10^8$ Cryopreserved CD34+ HSPCs were then thawed at 37 °C and cultured in supplemented cytokine rich SCGM media (CellGenix) containing recombinant cytokines at 100 ng/mL each Flt-3L, TPO, and SCF (PeproTech) and 35 nM UM171 (ExCellThera) in gas permeable vessels and incubated in 5% $CO_2$ + 5% $O_2$ for 48–72 h. The cells were then washed and resuspended in 3–10 mL of electroporation buffer (Hyclone). A GMP grade chemically modified single guide RNA (sgRNA) targeting the *HBB* locus was purchased from Agilent with modifications for 2'-O-methyl-3'-phosphorothioate at the three terminal nucleotides of the 5' and 3' ends with the sequence 5'-CTTGCCCCACAGGGCAGTAA-3'. The gene editing reagents were pre-complexed as an RNP containing 2 mg/mL sgRNA (Agilent Technologies) and 10 mg/mL SpyFi Cas9 (Aldevron) at a 2.5:1 molar ratio for 10 min at room temperature. Approximately 169 µL of RNP was added per 1 mL of cell suspension in electroporation buffer. For conditions testing the HDR booster, thawed i53 variant protein was mixed well by pipetting and added to the RNP at a concentration of 0.8 mg/mL (of total electroporation volume) following which the cells were electroporated using the MaxCyte GTx system using the CL1.1 or CL2 closed cartridge that are suitable for GMP manufacturing. Following electroporation, the cells were allowed to rest for 10 min in an incubator at 37 °C. In the meanwhile, prepared HBB-SNP virus carrying the corrected sequence for *HBB* was thawed and added at either $6.25 \times 10^2$ or $1.25 \times 10^3$ vector genomes/cell into culture media following which the electroporated cells were split equally (for different MOI conditions) and transferred to gas permeable culture vessels. At 16–24 h post-gene editing, the cells from each condition were collected and centrifuged at 300 x g for 10 min to pellet the cells. The supernatant was aspirated, and the cell pellet(s) were washed with and re-suspended in Plasma-Lyte buffer with 2% (v/v) HSA. A cell count was performed using the NC202 counter that uses AO/DAPI staining using the pre-set Cell count and Viability protocol. Cell counts were used to determine cell yield, viability and concentrations for cryopreservation. Following a final centrifugation step at 450 x g for 10 min, the cells were resuspended in cold cryopreservation media CryoStor CS5 (BioLife Technologies) and aliquoted into vials at a final concentration of $5 \times 10^6$ - $1.2 \times 10^7$ cells/mL. The vials were then subject to cryopreservation using a controlled rate freezer and storage in vapor phase $LN_2$ at $\leq$ -150 °C prior to performing all analytical metrics.

## Reporting summary
Further information on research design is available in the Nature Portfolio Reporting Summary linked to this article.

## Data availability
Atomic coordinates for crystal structures have been deposited in the Protein Data Bank (PDB) under accession codes: 8SVG, 8SVH, 8SVI, 8SVJ, and 8T2D. Mass spectrometry output data obtained from Jade Bio (used for additional validation of purified protein variants) is provided as Supplementary Data 1 (raw data is not accessible). Amplicon NGS (on and off-target *HBB* and i53 variant enrichment) and scRNA-seq data has been deposited in NCBI's Gene Expression Omnibus[52] and are accessible through GEO Series accession number GSE242757. The reference genome used alignments in sequencing-based assays was always GRCh38 (hg38; www.ncbi.nlm.nih.gov/datasets/genome/GCF_000001405.26/). Additional raw and processed data files are available from the authors upon request. Source data are provided with this paper.

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

## Acknowledgements

We thank Chris Bandoro for help with DNA Damage Response (DDR) assay development and lentiviral production troubleshooting, Ryan Rodriguez for guidance on optimizing lentiviral transduction of HSPCs, Dana Duan for assistance with lentiviral production, Jing Wei and the team at Jade Bio for mass spectrometry, and Ardian Wibowo, Joshua Carter, Anne Mulichak and Matt Kelker at Helix BioStructures for protein crystallization and X-ray crystallography data collection. We also would like to thank Akanksha Chhabra for expert advice on CD34+ cell type characterizations and prioritizations and Prof. Matt Porteus for his input on manuscript preparation.

## Author contributions

K.L.S. and J.A.P.B. designed experiments and wrote manuscript, with input from other authors. K.L.S. and O.E. designed and cloned constructs and generated lentivirus. K.L.S., O.E., W.M.M., G.M.C., B.J.S., and B.W. designed, developed, and implemented the pooled screening in HSPCs. J.A.P.B., W.M.M., K.A.H., C.D.L., and S.T. performed and interpreted NGS analyses for the pooled screens. J.K.H. and J.R.P. performed the purification of i53 proteins and biophysical analyses. J.K.H. and J.R.P. analyzed crystallographic data. K.L.S., J.A.P.B., O.E., S.K., E.C., T.L.G., B.J.Q., Y.G.S., M.V., K.A.W., and B.W. designed and carried out small scale HSPC editing and associated analyses. J.A.P.B., A.G., D.L., J.S., M.S., J.S.L., J.T., Y.Y., and P.L., and designed and carried out large scale HSPC editing and associated analyses. J.A.P.B., G.A., C.L.E., K.A.H., A.J., K.K., C.D.L., T.V. and S.T. performed and interpreted NGS analyses for editing at *HBB*. J.L.G., V.B.S., and D.P.D. provided project oversight. K.L.S. and J.R.P. supervised the study.

## Competing interests

All authors are current or former employees of Graphite Bio, Inc. and may own stock/options in the company. A patent application encompassing compositions of and methods for using the improved 53BP1 inhibitors has been filed with J.A.P.B., O.E., W.M.M., B.W., V.B.S., J.R.P. and K.L.S included as inventors and Graphite Bio, Inc. as the applicant (PCT/US2023/084675, pending).
