## [Peer Review File · Nature Communications]

Functional Screening in human HSPCs identifies optimized protein based enhancers of Homology Directed RepairREVIEWER COMMENTS

Reviewer #1 (Remarks to the Author):

Perez-Bermejo et al submit an interesting manuscript detailing discovery of novel engineered enhancers of homology directed repair, based upon rational mutagenesis screens of the engineered ubiquitin variant i53. Performing mutagenesis screens in primary human HSPCs alone is a difficult feat, and directly assessing HDR in these cells, which is baseline low efficiency, is an elegant method for discovering novel methods of improving HDR efficiency. The authors show how new i53 variants are predicted to alter binding strength of the i53 protein to 53BP1 through crystallography, and that these novel variants seem to further inhibit NHEJ thus promoting HDR without increasing markers of DNA damage response. There are nice mechanistic studies with amino acid granularity; great to see this type of investigation to hone in on how these variants are interacting with 53BP1 and why the variants may be favorable thermodynamically. The novel i53 variant L67R can also be used in combination with small molecule inhibitors of orthogonal NHEJ pathways to improve HDR ratios in HSPCs, which the authors show can further lead to reduction in the dose of genotoxic AAV HDR template delivery vector. This is really important for the gene edited therapy field as the authors' company has experienced a clinical setback possibly due to the use of AAV6 vectors in HSPCs in a clinical trial. They show versatility with increased HDR in HSPCs at multiple loci, with both small and large knockins. They also perform appropriate investigations of off-target effects and some safety profiling. The manuscript concludes by assessing i53-variant enhanced editing in a minimum clinical scale batch (~200e6 cells), which would be of great interest to the field as rarely can any academic or publicly-funded group afford such a costly manufacturing-scale experiment.

Overall the paper is very well written and contributes both a novel strategy for performing screens in HSPCs as well as a novel HDR enhancer with strong mechanistic studies to support future further rational design. However, there are two major discrepancies between the claims and the data supplied that need to be addressed in order to be fit for publication. An alternative hypothesis to fit the data is that the novel i53 protein increases the ratio of knockin cells by selectively killing off cells that have undergone NHEJ or MMEJ (perhaps by binding too strongly to 53BP1?). This would artificially increase the numerator in the HDR ratio at the expense of overall cell numbers, without increasing the yield of HDR+ cells at all. While this is -probably- not the case, the two major comments below suggest revisions that could dispel the alternative hypothesis and prove the authors' intended points. Namely, the authors need to supply data as to the actual number of cells input into the editing and output at the end of the process; they also need to account for template switching and PCR bias that is evident in their NGS data. These requested revisions should be doable without needing new experiments. Would just need additional existing data, or if not available or not provided would require significant modifications of the claims and reduce the overall impact. It is difficult to endorse the major claims of this manuscript otherwise.

MAJOR COMMENTS:

#1) Importantly, the point of the paper is to find a new method for increasing HDR in primary human HSPCs, in which it is notoriously difficult to successfully achieve HDR. Multiple prior attempts have been made to improve knockin in HSPCs, many cited by the authors, but as the authors point out both cell health and edited cell yield can suffer at the expense of techniques for improving HDR 'efficiency'. Here the authors specifically define HDR 'rates' as measured by the ratio of cells with successful HDR measured at the given timepoints (specifically 2 days post editing for NGS, 5 days post editing for flow cytometry). And while the authors show evidence of increased -ratio- of HDR as measured at those time points by either NGS or flow cytometry, none of their main figures, supplemental figures, nor accompanying text or captions gives any indication to the number of successfully edited cells. Not only does this omission limit the reproducibility of the study, it also calls into question if the HDR ratios reported are in fact of any practical benefit. Neither the measurement of DDR nor live cell gating by DAPI staining suffice to infer the number of alive cells post editing, they simply observe the sample of cells that happened to be surviving at the time of measurement. Further, even when the authors perform a very intriguing experiment and give the number of edited cells (~200e6), there is no further

mention of how many cells survive the process. The “% viability” in SuppFig5.1 does not suffice as it is again a ratio and not an absolute cell count. In other words, we do not know if the edited cells as assayed at that time point represent a pool of 100 surviving cells or 200,000,000 (or more). In fact, the authors do not disclose any of the smaller batch sizes of cells edited. The methods refer to cell concentrations and AAV concentration, but do not disclose how many cells were edited in a single electroporation well or cuvette; they also do not disclose the dose of RNP or the dose of the i53/i53 variant. This information should all be plainly stated for the reader as it should not be a trade secret. If this is indeed a methodology trade secret, the outcomes in terms of number of input cells and numbers of output cells (and what fraction are edited) should not be a secret. Certainly the field would be interested in 2x HDR in HSPCs but not if it also resulted in >2x less total cells, and this cannot be determined by the manuscript as written. Thus, it is impossible to judge if the authors have discovered an i53 variant that is actually beneficial to the field of genome editing and cell therapy.

#2) Many of the authors claims of increased HDR (and inhibited NHEJ) are based upon PCR of extracted cellular DNA and then NGS readouts (presumably at 2 days post editing per SuppFig 3 and methods). Assuming the PCR was not horribly biased by overamplification, then the NGS reads should reflect the alleles proportionately contributed by each cell. For example, a hypothetical 100 unique cells would contribute 200 unique alleles and each unique allele would be reflected by $\sim 1/200$ or 0.5% of the NGS reads. While almost all data is shown as relative increases in HDR compared to control seemingly $\sim 2x$ fold (Fig3C), the authors also nicely show that i53 variants can increase the % of GFP+ cells as measured by flow cytometry in Sup Figs 3.2 and 4.2. This protein based readout is not subject to PCR bias and confirms an increased ratio (though at what expense? See comment #1). Critically, in Supp Fig 3.2 A we are shown that the GFP+ population is baseline 26% as measured by flow cytometry. The gating may be too conservative and perhaps the GFP% is actually higher than stated, though similar numbers appear in SuppFig 3D at different loci. However, in SuppFig 3.2 B (the same cells as in 3A), the authors report that at baseline, 54% of alleles depict “HDR” as determined by NGS. This is not possible in primary human cells (diploid). If 26% of cells are GFP positive, meaning at least one allele has GFP inserted in frame, then those cells would contribute 13% of alleles (26 cells with 1 HDR allele = 26 HDR alleles out of 200, the remaining alleles being WT or with indels). If it was all biallelic editing, then the GFP+ cells would have two successful GFP insertions, one on each targeted allele, and 26 cells would contribute $26 \times 2 = 52$ out of 200 (26%) of alleles. Thus the depiction of 54% or greater alleles but only 26% protein level expression (even if more liberal gating is applied) implies one of two possibilities – either the authors have generated NGS reads from off target insertion (into HBD or another pseudogene), which should be obvious on the CRISPREsso readouts in the allele tables given that the authors display their adeptness in distinguishing repair off the HBD gene, or they are generating more HDR-appearing alleles during PCR than actually existed within the cell. The most classic reason for generating too many HDR-appearing alleles on sequencing of extracted DNA would be the problem of ‘template switching’ during PCR. Conceptually, this occurs when PCR amplicons initially generated by primer binding to genomic DNA have impartial extensions during early cycles – these impartial extensions on subsequent cycles bind to the delivered HDR template, which is in vast molar excess compared to genomic DNA (particularly with AAV delivery). Thus even with PCR primers outside of the homology arms, template switching can create the appearance of more alleles with perfect HDR than truly existed in the genomic DNA. While the colony PCRs in SuppFig 5.3 do perhaps mitigate the possibility, here it looks like even higher percentages of HDR and most are monoallelic. To address this, the authors should have done a control where they added a scrambled or off-target gRNA RNP plus the HDR template (by AAV), which would result in zero on-target HDR. In this control if any HDR is noticed in the allele, it would clearly be due to template switching. Interestingly the authors report in the methods (line 752) doing all the other controls (no editing, no template) but curiously not the off-target guide control. If this off-target guide control was indeed performed, it should be reported and the claims adjusted accordingly. If this off-target guide control was not performed, the authors should call out that they are overestimating the percentage of actual HDR by their PCR, but they can still assess the -qualitative- disappearance of certain alleles characteristic of NHEJ or MMEJ as in Fig 3 and SuppFig3.4 etc; without additional controls they cannot prove a fold increase in HDR as in lines 133 and 139. Again, no new experiments need be done, just

need to provide additional data, or alter the claims and stick to the relative differences as indicated by the protein based readouts. Depending on the results of Comment #1 (eg edited cell yield), they can still claim increasing successful HDR by inhibiting NHEJ but not at the quantitative level.

Additional Minor comments:

- 1) What is the total number of i53 protein variants tested in the lentiviral library? How many hits were found?
- 2) Its not clear that there is a callout to Fig 1E in the text.
- 3) Line 129 'selected i53 protein variants' – can you be clear in how these were selected? What was the rationale for eliminating others?
- 4) Line 132 and others – was IL2RG editing performed in XX or XY cells (or a mixture)? This would alter the efficiency of HDR.
- 5) Line 139 could you define what you mean by EC50
- 6) Figure 3B – was there a no protein control? Its possible I this is all artifact, that the variants increase editing over the WT i53, but don't actually have measurable benefit at the NGS level over leaving out i53 completely. This could be evidence for Major Comment #2.
- 7) At the end of the screening, were any i53 variants able to show reduced or abrogated function? Can you show some of that data in Fig 3 as a control?
- 8) Lines ~147-152 – How does the iindel spectrum change with the i53 protein but no HDR template? This would be the most likely way to visualize loss of NHEJ outcomes without the confounding bias of the HDR template (as in Major Comment #2).
- 9) Claim on line 162-163 i53 variants increase HDR by ONLY reducing NHEJ is not supported by the data (given Major comment #2).
- 10) Line 179, why aren't the -2 "longer edits" (is -2 a long edit) considered NHEJ if the +2 edits are considered NHEJ? Why aren't the -5 and -7 edits characterized as MMEJ? We don't have the full allele table to assess but my guess is there are not a lot of unique -5 and -7, and that instead there is a characteristic -5 or -7 deletion that would indicate MMEJ.
- 11) Lines 190-195 – while the quantity of off target was reduced, did the overall quality (ie spectrum of indels) also change at the off0-target sites?
- 12) IT is difficult to claim that these cells are phenotypically LT-HSCs without doing the mouse transplantation. Some of the authors have a preprint in Nature's Research Square showing long term repopulation by serial transplant, curiously with the same 4 HDRTs, so Im curious if this data actually exists but is being withheld. At best, we could call these HSPCs with surface markers of LT-HSCs (CD90+CD49f+) but there is not a true phenotype here.
- 13) Line 278 – should really be restated as increasing the HDR ratio rather than repair outcomes, as its still not clear what the absolute value of edited cells is after treatment with the novel i53 variants.
- 14) Line 564 – what is the dose of RNP? What is the number of cells edited in a batch? With so much other detail in the this manuscript, this omission seems too obvious.
- 15) Supp Fig 5.3B the y axis is labeled % colonies but the numbers indicate this is fraction of colonies.

Reviewer #2 (Remarks to the Author):

In this manuscript, Perez-Bermejo et al. explore the inhibition of 53BP1 to impede its recruitment to DNA damage sites, thereby activating homology-directed DNA repair (HDR) and enhancing the efficiency and safety of genome editing in human hematopoietic stem and progenitor cells (HPSCs) using CRISPR-Cas9. 53BP1 is a DNA damage response protein known to inhibit HDR. To inactivate 53BP1, the authors modified a previously reported engineered ubiquitin derivative (i53) that obstructs the chromatin-binding surface of the 53BP1 tandem Tudor domain (53BP1-Tudor). i53 had been obtained through screening a combinatorial library by another group. The authors employed targeted saturation mutagenesis at the i53-53BP1-Tudor interface in a pooled functional screening approach to identify i53 variants that enhance HDR. Through biophysical and X-ray crystallography studies, they demonstrated that an increased affinity of several i53 variants for 53BP1-Tudor, even if modest,

correlated with improved HDR efficiency. Subsequently, they characterized how the optimized i53 variants influenced the DNA repair outcome in HSPCs, showing that the variants increased HDR while reducing DNA repair by non-homologous end joining (NHEJ) but not by microhomology-mediated end joining (MMEJ). Under conditions approximating a therapeutic setting, they determined that i53 variants could enhance HDR without increasing genotoxicity. They also examined the impact of i53 variants on single-cell genotypes, noting, for instance, a ~30% increase in colonies bearing at least one HDR-corrected allele.

In the binding BLI assays presented in Supplemental Figure 1.6, it would be beneficial to include the values obtained for the on-rates, off-rates, and dissociation constants. Additionally, including a schematic of the TR-FRET assay would enhance the clarity of Supplemental Figure 1.6. Errors on IC₅₀s could also be indicated.

The legends for Supplemental Figure 1.6 do not correspond correctly to the panels. Legends for panels C, D and E are for D, E and F.

Overall, this is an interesting, comprehensive, and technically sound study that suggests blocking the chromatin recruitment of 53BP1 using a purified inhibitory protein may facilitate gene editing for therapeutic purposes.

Minor Points:

Some terms should be defined when first introduced in the text. For instance, "LT-HSC" (Long-Term Hematopoietic Stem Cell) is used in the main text but is first defined in the Methods section. Additionally, "MOI" should be defined upon first use.

Reviewer #3 (Remarks to the Author):

Perez-Bermejo JA et al. report on a very important limitation of the therapeutic application of genome editing technology: correcting genetic defects in hematopoietic stem and progenitor cells (HSPCs) without diminishing the cells' in vivo (post-transplant) long-term regenerative potential. The author describes a methodology to increase the frequency of HDR-modified alleles (therapeutic product) by specifically inhibiting the NHEJ DNA repair pathway in (HSPCs), using the CRISPR/Cas9-AAV6 platform. This study expands on an already engineered p53 inhibitor (WT to identify p53 variants that outperform the WT version in their ability to inhibit the recruitment of 53BP1 to double-strand breaks. To this end, the author developed a screening system to identify variants of i53 with higher affinity for 53BP1. Four candidates were selected and tested for their ability to increase HDR frequencies primarily at the HBB locus, though three additional genomic loci (HBA, CCR5, and IL2RG) were included.

The overall methodology used for screening and generating the i53 variants is sound. The screen was performed in HSPCs (clinically relevant cell type) and used HDR frequency as a functional readout to identify i53 variant candidates that outperform the i53 wild type.

Major concerns:

(A) While the i53 variants are more potent at inhibiting the NHEJ pathway compared to parental (WT) i53 protein

and seem to disrupt only the NHEJ pathway and not MMEJ, the increase in HDR between i53 WT and variants is incremental.

Fig. 3B i53 WT is ~%30 and i53 variants are barely reaching 40%

Fig. 3A Treatment with i53 (WT or variants) at high MOI decreases the overall % HDR compared to lower MOI. Is that due to virus-induced toxicity?

Sup Fig 3.2 D: the absolute number of HDR edited cells at the HBB locus with i53 WT vs variants is only incremental, and that's across all loci tested.

Sup Fig. 3.6 A: low or high MOI no difference; 3.6B: i53 WT: 58.9 HDR events vs 60.8, 61.1, 59.4 for i53 variants

In contrast to these i53 variants (or i53 WT) Winberger, S et al., Nature Comm 2023 showed AZD7648 compound (also tested in this study) to increase the HDR frequencies from ~20% (no treatment) to ~45%. With dual AZD + PolyQi reaching close to 80% HDR. Selvaraj, S. et al., Nat. Biotech, 2023 showed that AZD7648 can achieve a drastic increase in HDR [in HSC from ~ 15% (no treatment) to almost 40% (post-treatment)], even when a suboptimal sgRNA guide was used.

(B) Compared to recently published NHEJ (53BP1, DNAPKcs) and MMEJ (PolQ) inhibitors (Riesenbert S et al., Nat. Methods, 2022; Wimberger S et al., Nat. Comm, 2023; Selvaraj, S. et al., Nat. Biotech, 2023), the author shows that when combined with DNAPK inhibitor (AZD7648), i53 variants have locus dependent and additive effect. A strong argument was made against using AZD7648 in the clinic due to increased off-target activity (lines 263-267).

Was the dramatic increase (2% to 65%) in off-target activity seen across multiple loci? If not, or if they didn't test it, the author must remove this statement from the text. Alternately, the author could state that treatment with AZD during genome editing at the HBB locus increases off-target activity, and its clinical applications may be limited (e.g., not advised for sickle cell patients)

(C) The author claims that using i53 variants increases the frequency of HDR-modified LT-HSCs and early hematopoietic progenitors while decreasing the number of cells edited by the NHEJ pathway. Fig. 5A-C I can't tell if the i53 variants were better than the WT i53 (data missing).

Fig 5B Editing (RNP + AAV6) adds significant toxicity to the HSPC cells. If i53 variants are superior to i53 WT and other published inhibitory compounds, then the author should have presented side by side comparison of the absolute number of colonies obtained from WT (not editing), RNP only, RNP+AAV6 (no treatment), RNP +AAV6 (various inhibitors). Without this data it is difficult to assess the benefit (in terms of reducing toxicity) that i53 variants may have on LT-HSCs.

Fig. 5C (left panel, green bars) Same comment as for 5B. In addition, the middle bar (625 MOI + VHH variant): with an n=2 donor and large variation, one cannot conclude. If you remove the outlier (triangle in the middle bar), all look about the same (+/- VHH; high or low MOI). The claim "significant" (line 258) is unfounded. The author also states (line 258) that the frequencies of HDR modification persist in LT-HSC-derived colonies. The data in this manuscript does not support this conclusion. It takes two weeks to generate colonies, insufficient time to claim that the HDR is "persistent in the LT-HSCs." This can only be assessed in vivo through serial engraftment studies.

While the i53 variants do not show a dramatic improvement over i53 WT, at least not in vitro, they may outperform the WT i53 or other published inhibitors in vivo. Since most of the studies in this manuscript were performed at the HBB locus, to identify ways to improve the genome editing strategy for treating sickle cell patients, testing these variants in vivo, through xenotransplantation studies would have been necessary and in line with the scope of this study. Given the recent publications, it would also increase the novelty of the study.

Reducing the time of ex vivo culture and manipulation (from 5 days to 4 or 3) is another critical parameter in preserving the cells' engraftment potential. An additional interesting experiment would be to see if treating CD34+ at day 2 (rather than day 3) of ex vivo expansion increases the frequency of HDR by day 3 or, at most, day 4 (instead of day 5). If it does, how would these variants compare to other published inhibitors? NHEJ is active throughout cell cycle stages, while HDR machinery is only active at the S-G2 phases. If the cells are treated on day 2 (the time when the cells complete the G1 phase and enter the S phase) and allowed to cycle for only an additional day, would a high HDR frequency be observed while increasing the cells' regenerative potential (CFU and engraftment)?

RESPONSE TO COMMENTS FOR RESUBMISSION PEREZ-BERMEJO ET AL, 2023 (NATURE COMMUNICATIONS NCOMMS-23-36442-T)

We thank the editor and all the reviewers for carefully reading and reviewing our manuscript and providing us with their insightful and helpful comments. Though the SCD clinical and the HDR-based research programs were terminated at Graphite Bio in early 2023, we are thrilled to have this exciting opportunity to share the details of our work through publication in Nature Communications. We are hopeful that by publishing this work, others in the field can leverage the functional screening system we built in order to continue the identification and improvement of protein-based HDR enhancers and that the new i53 variants identified in this study can be broadly utilized by the field to better understand the impact of 53BP1 inhibition on different HDR-based gene therapy approaches. We very much appreciate the interest in our work and the positive and constructive feedback expressed by the reviewers.

We have included our point-by-point response to the comments below and highlighted (using 'track changes') the new data and text that we have incorporated to enhance the quality of our revised manuscript. As a general principle, in multiple comments the reviewers requested data that we readily had and perhaps because of the over familiarity with the data and system, we overlooked the importance of including it. We appreciate the reviewers' efforts in helping us refine our manuscript; by addressing their concerns and suggestions, we believe the current manuscript is much improved as a result.

Raw and processed data from genomics experiments is now available on GEO using the following information:

Link: <https://www.ncbi.nlm.nih.gov/geo/query/acc.cgi?acc=GSE242757>

Reviewer token (password): wbgnmokohnyvlgf

POINT-BY-POINT RESPONSE TO REVIEWER COMMENTS

Reviewer #1 (Remarks to the Author):

Perez-Bermejo et al submit an interesting manuscript detailing discovery of novel engineered enhancers of homology directed repair, based upon rational mutagenesis screens of the engineered ubiquitin variant i53. Performing mutagenesis screens in primary human HSPCs alone is a difficult feat, and directly assessing HDR in these cells, which is baseline low efficiency, is an elegant method for discovering novel methods of improving HDR efficiency. The authors show how new i53 variants are predicted to alter binding strength of the i53 protein to 53BP1 through crystallography, and that these novel variants seem to further inhibit NHEJ thus promoting HDR without increasing markers of DNA damage response. There are nice mechanistic studies with amino acid granularity; great to see this type of investigation to hone in on how these variants are interacting with 53BP1 and why the variants may be favorable thermodynamically. The novel i53 variant L67R can also be used in combination with small molecule inhibitors of orthogonal NHEJ pathways to improve HDR ratios in HSPCs, which the authors show can further lead to reduction in the dose of genotoxic AAV HDR template delivery vector. This is really important for the gene edited therapy field as the authors' company has experienced a clinical setback possibly due to the use of AAV6 vectors in HSPCs in a clinical trial. They show versatility with increased HDR in HSPCs at multiple loci, with both small and large knockins. They also perform appropriate investigations of off-target effects and some safety profiling. The manuscript concludes by assessing i53-variant enhanced editing in a minimum clinical scale batch (~200e6 cells), which would be of great interest to the field as rarely can any academic or publicly-funded group afford such a costly manufacturing-scale experiment.

Overall the paper is very well written and contributes both a novel strategy for performing screens in HSPCs as well as a novel HDR enhancer with strong mechanistic studies to support future further rational design. However, there are two major discrepancies between the claims and the data supplied that need to be addressed in order to be fit for publication. An alternative hypothesis to fit the data is that the novel i53 protein increases the ratio of knockin cells by selectively killing off cells that have undergone NHEJ or MMEJ (perhaps by binding too strongly to 53BP1?). This would artificially increase the numerator in the HDR ratio at the expense of overall cell numbers, without increasing the yield of HDR+ cells at all. While this is -probably- not the case, the two major comments below suggest revisions that could dispel the alternative hypothesis and prove the authors' intended points. Namely, the authors need to supply data as to the actual number of cells input into the editing and output at the end of the process; they also need to account for template switching and PCR bias that is evident in their NGS data. These requested revisions should be doable without needing new experiments. Would just need additional existing data, or if not available or not provided would require significant modifications of the claims and reduce the overall impact. It is difficult to endorse the major claims of this manuscript otherwise.

MAJOR COMMENTS:

#1) Importantly, the point of the paper is to find a new method for increasing HDR in primary human HSPCs, in which it is notoriously difficult to successfully achieve HDR. Multiple prior attempts have been made to improve knockin in HSPCs, many cited by the authors, but as the authors point out both cell health and edited cell yield can suffer at the expense of techniques for improving HDR 'efficiency'. Here the authors specifically define HDR 'rates' as measured by the ratio of cells with successful HDR measured at the given timepoints (specifically 2 days post editing for NGS, 5 days post editing for flow cytometry). And while the authors show evidence of increased -ratio- of HDR as measured at those time points by either NGS or flow cytometry, **none of their main figures, supplemental figures, nor accompanying text or captions gives any indication to the number of successfully edited cells.** Not only does this omission limit the reproducibility of the study, it also calls into question if the HDR ratios reported are in fact of any practical benefit. Neither the measurement of DDR nor live cell gating by DAPI staining suffice to infer the number of alive cells post editing, they simply observe the sample of cells that happened to be surviving at the time of measurement. Further, even when the authors perform a very intriguing experiment and give the number of edited cells (~200e6), **there is no further mention of how many cells survive the process. The "% viability" in SuppFig5.1 does not suffice as it is again a ratio and not an absolute cell count.** In other words, we do not know if the edited cells as assayed at that time point represent a pool of 100 surviving cells or 200,000,000 (or more).

Thank you for highlighting the importance of this aspect of improving HDR—not just increasing the frequency of HDR but also maintaining (or improving) cell numbers. In the revision we have added information on percent recovery of cells in our 200M editing experiment to Supplementary Fig 5.1 and have included details in the figure caption on how percent recovery was calculated (by counting the harvested viable cells pre-cryopreservation relative to viable cell numbers counted prior to electroporation). We have also added text to reflect our conclusions from this important data: that the addition of the i53 variant has no significant impact (negative or positive) on cell recovery post editing process at this scale (line 222).

We've also added cell recovery numbers for the small-scale experiment shown in Figure 4C and Supplementary Figure 4.4 probing the differential impact of 53BP1 and DNAPK inhibition across HBB-SNP AAV6 MOIs. Relative cell recovery numbers were calculated by dividing the total number of cells counted at cell harvest for each condition relative to an untreated control. We observed no significant

impact (negative or positive) on cell recovery numbers when editing cells using small-scale conditions incorporating the i53 variant (L67R) or DNAPKi (AZD7648) additives alone relative to the no additive control. When re-examining this important data, however, we did notice a slight, but significant, reduction in recovery when using both inhibitors together, perhaps hinting at some toxicity with dual 53BP1 and DNAPKi inhibition in HSPCs. We have noted this observation in the figure caption.

Additionally, we have included new data into Supplementary Figure 3.5 addressing the impact of i53 variant concentration on apoptosis levels within edited cell populations. We believe the inclusion of this data provides additional support for our claim that incorporating i53 variants (or increasing doses of i53 variants) to our HSPC ex vivo editing protocol has no significant negative impact on the number of healthy, dead, apoptotic cells within an edited cell population. The method for the apoptosis analysis, which is flow-based and does not pre-sort for live cells, is added at line 1295. Representative gating is shown in Supplementary Figure 3.5.

Lastly, we would like to point out that we did not observe an effect of i53 variant usage on P21 expression (as seen in figures S3.8E, S4.4E, and S5.1D), which is a reporter for not only DNA damage response but also overall cell stress. The decrease in P21 signal observed upon using DNAPKi small molecules has also been reported by others (Selvaraj et al, 2023) and we attribute it mostly to the direct interplay between P21 and the DNAPK pathway.

Ultimately, though the use of HDR enhancing additives did not result in reduced cell toxicity *per se*, the reduction in homology repair template (i.e. AAV) that is enabled by these does result in a reduction of cellular stress (the correlation between AAV template and cell stress has been documented by others, for example Xu et al, 2023).

In fact, the authors do not disclose any of the smaller batch sizes of cells edited.

Our apologies for this omission. We have added information about the scale of each experiment by providing the number of cells per electroporation cuvette for each small scale editing experiment into the figure captions. We have also indicated if the cells were split across conditions (different AAVs, different MOIs, and/or +/- DNAPKi) post electroporation.

The methods refer to cell concentrations and AAV concentration, but do not disclose how many cells were edited in a single electroporation well or cuvette; they also do not disclose the dose of RNP or the dose of the i53/i53 variant.

Thank you for the interest in these critical details of the experimental protocol. We have added to our methods a table (Table M1, line 1010) that lists the range of cells in the electroporation cuvettes used for the small scale editing experiments detailed in the manuscript. We have also added information on the final concentrations of sgRNA and Cas9 (precomplexed as RNP) used in these cuvettes (lines 993-994). The i53 variant concentrations are stated for each experiment in the corresponding figure captions. The range of concentrations used throughout the study is listed in the methods (lines 1001-1003). Additionally, Figure 3C highlights our suggested working concentrations for the variants identified from the screen (0.8 mg/mL). We appreciate the suggestions and hope that this will allow readers to more readily assess the procedures and others to more reliably reproduce the work described.

This information should all be plainly stated for the reader as it should not be a trade secret. If this is indeed a methodology trade secret, the outcomes in terms of number of input cells and numbers of

output cells (and what fraction are edited) should not be a secret. Certainly the field would be interested in 2x HDR in HSPCs but not if it also resulted in >2x less total cells, and this cannot be determined by the manuscript as written. Thus, it is impossible to judge if the authors have discovered an i53 variant that is actually beneficial to the field of genome editing and cell therapy.

We thank this reviewer for pointing out these areas for improvement by adding additional information in the manuscript. We hope that by including the additional cell recovery, apoptosis, editing protocol, and cell number information detailed above, we have provided more robust description of the use and utility of the screening hits to the field. In sum, the increase in HDR did not come with a cost in decreased in cell number. We have added a sentence in the discussion to highlight this finding.

#2) Many of the authors claims of increased HDR (and inhibited NHEJ) are based upon PCR of extracted cellular DNA and then NGS readouts (presumably at 2 days post editing per SuppFig 3 and methods). Assuming the PCR was not horribly biased by overamplification, then the NGS reads should reflect the alleles proportionately contributed by each cell. For example, a hypothetical 100 unique cells would contribute 200 unique alleles and each unique allele would be reflected by $\sim 1/200$ or 0.5% of the NGS reads. While almost all data is shown as relative increases in HDR compared to control seemingly $\sim 2x$ fold (Fig3C), the authors also nicely show that i53 variants can increase the % of GFP+ cells as measured by flow cytometry in Sup Figs 3.2 and 4.2. This protein based readout is not subject to PCR bias and confirms an increased ratio (though at what expense? See comment #1). Critically, **in Supp Fig 3.2 A we are shown that the GFP+ population is baseline 26% as measured by flow cytometry. The gating may be too conservative and perhaps the GFP% is actually higher than stated, though similar numbers appear in SuppFig 3D at different loci. However, in SuppFig 3.2 B (the same cells as in 3A), the authors report that at baseline, 54% of alleles depict “HDR” as determined by NGS. This is not possible in primary human cells (diploid).** If 26% of cells are GFP positive, meaning at least one allele has GFP inserted in frame, then those cells would contribute 13% of alleles (26 cells with 1 HDR allele = 26 HDR alleles out of 200, the remaining alleles being WT or with indels). If it was all biallelic editing, then the GFP+ cells would have two successful GFP insertions, one on each targeted allele, and 26 cells would contribute $26 \times 2 = 52$ out of 200 (26%) of alleles. Thus the depiction of 54% or greater alleles but only 26% protein level expression (even if more liberal gating is applied) **implies one of two possibilities – either the authors have generated NGS reads from off target insertion (into HBD or another pseudogene),** which should be obvious on the CRISPREsso readouts in the allele tables given that the authors display their adeptness in distinguishing repair off the HBD gene, **or they are generating more HDR-appearing alleles during PCR than actually existed within the cell. The most classic reason for generating too many HDR-appearing alleles on sequencing of extracted DNA would be the problem of ‘template switching’ during PCR.** Conceptually, this occurs when PCR amplicons initially generated by primer binding to genomic DNA have impartial extensions during early cycles – these impartial extensions on subsequent cycles bind to the delivered HDR template, which is in vast molar excess compared to genomic DNA (particularly with AAV delivery). Thus even with PCR primers outside of the homology arms, template switching can create the appearance of more alleles with perfect HDR than truly existed in the genomic DNA. While the colony PCRs in SuppFig 5.3 do perhaps mitigate the possibility, here it looks like even higher percentages of HDR and most are monoallelic. To address this, **the authors should have done a control where they added a scrambled or off-target gRNA RNP plus the HDR template (by AAV), which would result in zero on-target HDR.** In this control if any HDR is noticed in the allele, it would clearly be due to template switching. Interestingly the authors report in the methods (line 752) doing all the other controls (no editing, no template) but curiously not the off-target guide control. If this off-target guide control was indeed performed, it should be reported and the claims adjusted accordingly. **If this off-target guide control was not performed, the**

authors should call out that they are overestimating the percentage of actual HDR by their PCR, but they can still assess the -qualitative- disappearance of certain alleles characteristic of NHEJ or MMEJ as in Fig 3 and SuppFig3.4 etc; without additional controls they cannot prove a fold increase in HDR as in lines 133 and 139.

We would like to thank this reviewer for these insightful comments and for taking valuable time to provide a clear explanation of how PCR can result in over-estimation by template switching during PCR. We agree that template switching has been an issue in many studies that provide measures of genomic engineering events. We also strongly support the need to be careful about independently validating the assay using other measures such as single cell analysis and flow cytometry and appreciate the reviewer having this understanding. We also 100% agree with the math about the relationship between “allele” targeting, “cell” targeting and the relative proportion of monoallelic and bi-allelically targeted cells and again thank the reviewer for spending the time to work through that explanation.

The cells in Supp3.2A (now Supp3.2C) and cell in Supp3.2B (now Supp3.2D) are not the same cells; they have been edited in parallel but treated with different AAV6s and have been analyzed differently (HBB-UBC-GFP AAV6-treated cells with flow analysis only; HBB-SNP AAV6-treated cells with NGS analysis only). To better clarify, we have added a label designating the different AAV6s to Figure Supp3.2, and a schematic on Supplementary Figure 3.1 B-C).

The design of both the HBB-SNP and HBB-UBC-GFP AAV6 has been previously described in the literature (Dever et. al. Nature 2016; Vakulskas et. al. Nat Med 2018; Wilkinson et. al. Nat Comm 2021, as cited in the manuscript). However, it is clear from the reviewer’s comments and major concerns that we had not clearly outlined the differences between these two HDR donors, as well as the expected differential impact of their use on %HDR, in our submitted manuscript. To help address this, we have added a schematic detailing the design of the HBB-targeted AAV constructs (Supplementary Figure 3.1 B-C). Though these two AAVs target the same location and leverage the use of the same gRNA, the baseline HDR efficiency in cells edited with the two HBB AAV-supplied templates should differ at least in part due to differences in the knock-in size - integration of the UBC-GFP cDNA in the HBB-UBC-GFP AAV6 construct requires a knock-in of ~2.2 kB to enable GFP protein expression (vs. no net insertion in the case of the HBB-SNP donor).

Although the HBB-SNP and HBB-UCB-GFP assays are reporting on an intrinsically different editing strategy, there is a very clear linear correlation between them (now shown in Supplementary Figure 3.1D). The HBB-SNP NGS readout provides a more complete picture of the editing outcomes and also utilizes a strategy utilized for therapeutic genome editing. However, given the correlation between the two assays, in this report we leverage the use of the HBB-UBC-GFP strategy to provide a quicker and scalable readout. Importantly, the GFP knock-in strategy allows us to study HDR outcomes in other loci for which an NGS has not been developed (such as *HBA*, *IL2RG* and *CCR5* in this study). We note that both NGS, by measuring allele percentage in the population, and flow cytometry, which measures cell knock-in percentage, are important to measure and report.

Though clarification of the different AAV6 donors should help elucidate the differences in the baseline HDR observed by the reviewer when comparing %HDR (NGS) and %GFP shown in Supp3.1 and Supp3.2, it is also clear through the reviewer’s response that we need to strengthen the manuscript’s quantitative claims associated with the NGS-based assay. When analyzing cells edited with HBB-SNP AAV6, we use an NGS assay that leverages a previously described and validated in/out PCR to circumvent background reads from the AAV repair templates (i.e. by using primers that discriminate

between *HBB* and paralog gene sequences, and that are located outside the homology arm sequence region to avoid template amplification) (Dever et. al. Nature 2016; Vakulskas et. al. Nat Med 2018; Wilkinson et. al. Nat Comm 2021; Lattazani et. al. Sci. Transl. Med. 2021, all referenced in-text). However we acknowledge, as the reviewer highlights, that this in-out strategy does not completely preclude the possibility of template switching. To add additional information related to this issue, we have added NGS results when analyzing editing under different control conditions, including a “no AAV” condition (where we obtain <<1% positive reads), and an “AAV but no RNP” condition (where we observe no amplification from the AAV template itself or see significant problems with template switching) (see Supplemental Figure 3.1). Of note, HBD outcome reads remain very low in these controls even though our classification of NGS outcomes as HBD is quite conservative -we assign to this category every read that contains any number of SNPs tracking to the *HBD* gene, and there are no ‘substitution’ hits in CRISPResso other than the ones that track to this locus. No ‘off-target gRNA’ control is included in this study, but we believe that the lack of off-target amplification is proven by the combination of the ‘no AAV’ and ‘no RNP’ controls and the fact that the amplicon sequencing primers are located in the *HBB* locus outside the homology arm region. In particular, the ‘no RNP’ control is roughly equivalent to ‘scrambled RNP’ in this context, as it is a condition where no on-target breaks are generated. In addition, we have included a panel on the *HBB*-SNP NGS readout when using decreasing amounts of AAV template (Supplemental Figure 3.1), and where we can observe that the assay can effectively capture quantitative changes in template incorporation. This amplicon sequencing assay was also cleared by the FDA as a qualifying assay for the clinical product at Graphite Bio, providing additional confidence that our NGS analysis is capable of providing quantitative readouts for HDR efficiencies.

Again, no new experiments need be done, just need to provide additional data, or alter the claims and stick to the relative differences as indicated by the protein based readouts. **Depending on the results of Comment #1 (eg edited cell yield), they can still claim increasing successful HDR by inhibiting NHEJ but not at the quantitative level.**

We appreciate the reviewer providing guidance that no new experiments need to be done but that additional information is needed from the existing experiments. Given the discussion of the points above, and the additional data we are now able to provide about cell yields and NGS assay results under different controls, we believe that our study is able to provide quantitative conclusions about HDR increase/NHEJ reduction. We greatly appreciate this reviewer’s comments and concerns, and we believe they have contributed to make this manuscript’s claims stronger. Please do not hesitate to get back to us through the editorial system if there are critical concerns remaining.

Additional Minor comments:

1) What is the total number of i53 protein variants tested in the lentiviral library? How many hits were found?

Thank you for the suggestion - we have now included the number of variants tested and hits for each library in the captions of Figure 1 and Supplemental Figures 1.3 and 1.4. We observed approximately 5-10% positive hits in each of the libraries screened. Our approach to i53 variant optimization was to use the screen to identify and prioritize promising variants, which then get validated as purified protein, as shown in subsequent figures.

2) Its not clear that there is a callout to Fig 1E in the text.

Thank you for bringing this to our attention - we have added a callout to Fig. 1E on line 116 (as this line discusses the trends observed in the data) and have moved the callout to Fig. 1D and S1.4 to the

previous line (line 114, as this line discusses the details of our second round of screening building on the hits from the first round of screening- L67R and L67H).

3) Line 129 ‘selected i53 protein variants’ – can you be clear in how these were selected? What was the rationale for eliminating others?

We selected top variants from each screen (selected using data from “validation screens,” shown in Supp Fig 1.5). Initial tests with a number of hits from the first round of screening (varying residues 67 and 68, Supp Fig 1.3) suggested a functional equivalency of the top hits when incorporated as protein-based additives. To streamline our more intensive characterizations of editing and cell health outcomes when using library hits as HSPC editing protein-based additive, we moved forward with primarily using the top variants from the second round of screening (T12Y.T14E.L67R and T12V.T14H.L67H) and their parental hits from the first round of screening (L67R and L67H). To clarify this, we have replaced the wording “selected variants” with “top variants from each round of screening validation” (line 158).

4) Line 132 and others – was IL2RG editing performed in XX or XY cells (or a mixture)? This would alter the efficiency of HDR.

All IL2RG editing was done in XY cells (donor info by provider). We have added this information to the relevant figure captions. As we and the reviewer recognize, using a target on the X-chromosome in XY cells, collapses the potential discrepancy between allele targeting and cell targeting. They should be identical.

5) Line 139 could you define what you mean by EC50

We defined EC50 as the concentration of i53 variant required for half of the maximum increase in HDR that can be obtained when using the additive. Similarly, we use IC50 when referring to the concentration of i53 variant required for half of the maximum reduction in NHEJ. A line clarifying how these values are defined has been added to the legend of main Figure 3. Since these are measures of potency, we also changed the main text slightly to make it clearer (line 172).

6) Figure 3B – was there a no protein control? Its possible I this is all artifact, that the variants increase editing over the WT i53, but don’t actually have measurable benefit at the NGS level over leaving out i53 completely. This could be evidence for Major Comment #2.

Yes, there were cells from each donor edited with no protein additive controls; the average HDR in these cells is depicted by the dotted line and is designated by the “no protein” label. For a more comprehensive analysis of HDR in cells edited with the different i53 variants as compared to the no protein controls, we have added a new Supplementary Figure (3.4). The addition of all i53 proteins (with the exception of i53 WT at 0.2 mg/mL) lead to a significant improvement in %HDR (NGS analysis, $p\text{-val} < 0.0001$) relative to the no protein additive control.

7) At the end of the screening, were any i53 variants able to show reduced or abrogated function? Can you show some of that data in Fig 3 as a control?

We have included new data using a purified negative control from our screening system, i53-DM (shown in SuppFig1.2B, 1.2C, 1.5C, 1.5D, and 1.5E), into a Supplementary Figure (3.2). The addition of this negative protein control has no effect on HDR when using either HBB-UBC-GFP or HBB-SNP AAV6 as HDR templates (i.e. not significantly different that the addition of no protein). Aside from i53-DM, we only expressed and purified top “hit” variants from the screen (i.e. those shown in Supp 3.2C and D); all variants tested exhibited improved function over i53 WT.

8) Lines ~147-152 – How does the indel spectrum change with the i53 protein but no HDR

template? This would be the most likely way to visualize loss of NHEJ outcomes without the confounding bias of the HDR template (as in Major Comment #2).

We agree with this reviewer that the study of the effects of i53 variants in an “RNP only” (no template) condition would be valuable for the interpretation of the impact of i53 variants in editing outcomes. This data was indeed present in the original version of the manuscript, but it was not contextualized properly – supplemental Figure 4.4 includes an “MOI = 0” condition, and displays a clear reduction in the NHEJ outcomes (about 50%) for both i53 variant L67R and DNAPKi small molecules, and an even larger decrease when both are combined. Of note, in this condition only the NHEJ inhibition promoted a clear increase in MMEJ and HBD outcomes, which is not observed when an AAV template is present (presumably because AAV competes for other homology-based repair templates).

We have introduced a short comment on this point in lines 480-483, where the data is first introduced. In addition, the newly added assay QC data on Supplemental Figure 3.1 includes a ‘no AAV’ condition which clearly shows that the lack of AAV HDR signal when the repair template is absent.

Interestingly, the increase in HBD repair outcome in ‘no AAV’ conditions also hint at an alternative therapeutic gene editing strategy where the pathogenic *HBB* SNP could be replaced by the homologous HBD sequence without providing any exogenous template sequence, though the frequency we observed in this experiment would be too low for therapeutic use currently. The phenomenon of HBD being used as the donor template is similar to how the classic DR-GFP assay to measure HDR developed in the Jasin lab works. Although in the DR-GFP assay the donor template is closer to the break and is 100% sequence identical to the target where the break is induced.

9) Claim on line 162-163 i53 variants increase HDR by ONLY reducing NHEJ is not supported by the data (given Major comment #2).

As discussed extensively above and in light of the added additional data surrounding the NGS assay qualification, we believe that our study is now able to better provide quantitative conclusions about HDR increase and NHEJ reduction. We agree with the reviewer that the mechanism by which HDR is increased and NHEJ is decreased using i53 is probably more complex than when simply using a small molecule inhibitor of DNA-PKcs which acts to simply inhibit canonical NHEJ. To reflect this, claims about mechanism have been adjusted so we don’t claim that the mechanism is solely NHEJ inhibition, and to reflect that 53BP1 is more of an HDR inhibitor than an active NHEJ mediator. We would like to thank the reviewers for their valuable input in this regard, and we believe the manuscript more accurately represent the function of 53BP1 and its inhibitors as we know them.

10) Line 179, why aren’t the -2 “longer edits” (is -2 a long edit) considered NHEJ if the +2 edits are considered NHEJ? Why aren’t the -5 and -7 edits characterized as MMEJ? We don’t have the full allele table to assess but my guess is there are not a lot of unique -5 and -7, and that instead there is a characteristic -5 or -7 deletion that would indicate MMEJ.

This reviewer is correct that the classification of editing outcomes was indeed not arbitrary, but empirical. To assign INDELS as MMEJ-mediated, we surveyed the impact of a *POLQ* (a key MMEJ protein) knockdown on unique INDELS at *HBB* (detailed in SuppFig3.4, referenced in line 170). Every other INDEL was classified as NHEJ. It is correct that there are prominent deletions ranging from –8 to –11 that were impacted by this *POLQ* knockdown and were thus designated MMEJ (interestingly, not all unique deletions with the same length were associated with the same editing pathway). When you compare SuppFig3.4D (INDELS affected by MMEJ knockdown) to SuppFig3.5A (indels affected by 53BP1 inhibition) and SuppFig4.3A (indels affected by DNAPKi inhibition) one can see a satisfying orthogonal impact on NHEJ and MMEJ inhibition on the individual indels at *HBB*; those that were designated as MMEJ are clearly not impacted by either 53BP1 nor DNAPK inhibition.

As for the classification of edits as “long” or “short”, this distinction was made in the context of comparing editing outcomes affected by DNAPKcs or 53BP1 inhibitors (Supplemental Figure 4.3B). We observed that the former was more effective at downregulating net -1, +1, +2 INDELS, whereas i53 variants were more effective at downregulating INDELS of other lengths. We apologize for the admittedly confusing nomenclature, and have modified the text and figures to use a clearer, more explicit way of classifying them (“-1, +1, +2” vs “other INDELS”).

To make this designation clearer, we have added Supplementary Table 3.1 that better clarifies the unique repair outcomes at HBB and their contribution to the editing profile. In addition, Supplementary Table 3.2 outlines the different outcomes associated to HBD recombination (by sequence homology to this locus). We have also introduced modifications to the main text to make this classification easier.

11) Lines 190-195 – while the quantity of off target was reduced, did the overall quality (ie spectrum of indels) also change at the off-target sites?

We would like to thank this reviewer for their question, which prompted us to look more in depth at the identity of the INDELS at the off-target site. It is indeed tempting to speculate that the use of inhibitors could promote editing outcomes that are not observed at all when no additive is used. We have now added a new panel, Supplemental Figure 4.5, which outlines the INDELS at this site and how they are affected by the use of inhibitors (DNAPKi or i53 variant L67R). The analysis was challenging due to the low % editing in this off target site (1-2% reads overall), but it shows that the most prevalent edit is a 9 nucleotide deletion, which is increased when using the DNAPKi small molecule either alone or in combination with the i53 variant. Given the OT1 site is very similar to the on-target site in the sequence that generates the -9 deletion, it is not surprising that we see the identical -9 deletion at both sites. From the work at the on-target site, this -9 deletion is most likely result of MMEJ. We hypothesize this edit increases for the lack of a better homology repair template, similar to what we observe when we do “no AAV/RNPonly” editing (Figure S4.4B) and as per our observation that increasing AAV MOI decreases MMEJ (Figure S3.1F and many other panels).

12) IT is difficult to claim that these cells are phenotypically LT-HSCs without doing the mouse transplantation. Some of the authors have a preprint in Nature’s Research Square showing long term repopulation by serial transplant, curiously with the same 4 HDRTs, so Im curious if this data actually exists but is being withheld. At best, we could call these HSPCs with surface markers of LT-HSCs (CD90+CD49f+) but there is not a true phenotype here.

The Nature Research Square preprint this reviewer refers to (Baik et al, 2023) describes work completed in the Porteus lab at Stanford and is not associated with the work presented here. In particular, Dr Danny Dever is listed in that preprint as affiliated to Graphite Bio because that was his occupation at the time the preprint was published, but not when the work described in the article was performed. We apologize for the confusion this might have caused.

We agree with the reviewer that the use of the term “phenotypic LT-HSC” is not supported by in-house experimental data, instead inferred from previous literature research and prior experience of some of the authors and scientific advisors. We were very interested in probing the impact of these variants in a mouse transplantation study, but we were not able to complete these studies. We were still very interested in the potential impact of the i53 variants on editing outcomes in the LT-HSC subpopulation and thus speculated we could leverage an extensive panel to enrich for and investigate editing outcomes in these cells defined phenotypically by cell surface markers (we are the standard definition of CD34+CD45RA-CD90+CD201+CD49f+CD49c+). To better represent this limitation of the study

and our lack of a direct mouse engraftment study, we have gone ahead and modified all the references to "HSPCs with surface markers associated with LT-HSCs".

13) Line 278 – should really be restated as increasing the HDR ratio rather than repair outcomes, as its still not clear what the absolute value of edited cells is after treatment with the novel i53 variants.

As discussed above, we have added %recovery values to Supp Fig 5.1, which should demonstrate we are increasing HDR repair outcomes in edited cell pools without negatively impacting cell viability.

14) Line 564 – what is the dose of RNP? What is the number of cells edited in a batch? With so much other detail in the this manuscript, this omission seems too obvious.

We apologize for the oversight in not including these important details. We have now updated the methods section with this information, including a table with cell numbers used in different electroporation formats (Table M1) and an in-text description of the final concentration of Cas9 and gRNA used (in electroporation cuvette: 0.45 mg/mL Cas9, 0.24 ug/uL sgRNA).

15) Supp Fig 5.3B the y axis is labeled % colonies but the numbers indicate this is fraction of colonies.

Our mistake - the numbers on y axis have been changed to reflect the %colonies label and the numbers represented in the graphs.

Reviewer #2 (Remarks to the Author):

In this manuscript, Perez-Bermejo et al. explore the inhibition of 53BP1 to impede its recruitment to DNA damage sites, thereby activating homology-directed DNA repair (HDR) and enhancing the efficiency and safety of genome editing in human hematopoietic stem and progenitor cells (HPSCs) using CRISPR-Cas9. 53BP1 is a DNA damage response protein known to inhibit HDR. To inactivate 53BP1, the authors modified a previously reported engineered ubiquitin derivative (i53) that obstructs the chromatin-binding surface of the 53BP1 tandem Tudor domain (53BP1-Tudor). i53 had been obtained through screening a combinatorial library by another group. The authors employed targeted saturation mutagenesis at the i53-53BP1-Tudor interface in a pooled functional screening approach to identify i53 variants that enhance HDR. Through biophysical and X-ray crystallography studies, they demonstrated that an increased affinity of several i53 variants for 53BP1-Tudor, even if modest, correlated with improved HDR efficiency. Subsequently, they characterized how the optimized i53 variants influenced the DNA repair outcome in HSPCs, showing that the variants increased HDR while reducing DNA repair by non-homologous end joining (NHEJ) but not by microhomology-mediated end joining (MMEJ). Under conditions approximating a therapeutic setting, they determined that i53 variants could enhance HDR without increasing genotoxicity. They also examined the impact of i53 variants on single-cell genotypes, noting, for instance, a ~30% increase in colonies bearing at least one HDR-corrected allele.

- In the binding BLI assays presented in Supplemental Figure 1.6, it would be beneficial to include the values obtained for the on-rates, off-rates, and dissociation constants.** We have included a data table detailing the values obtained in the binding BLI assays (Supplementary Table 1.1).
- Additionally, including a schematic of the TR-FRET assay would enhance the clarity of Supplemental Figure 1.6.** A schematic has been added (Supp Fig 1.6E).

- **Errors on IC₅₀s could also be indicated.** We apologize for the omission – the table has now been updated with values ± Standard Error of the Mean. We have also replaced “IC₅₀” with “EC₅₀”, as it seemed more appropriate (since we are not calculating direct inhibition).
- **The legends for Supplemental Figure 1.6 do not correspond correctly to the panels. Legends for panels C, D and E are for D, E and F.** Our mistake - this has been corrected, and updated to reflect the addition of the new TR-FRET schematic panel.

Overall, this is an interesting, comprehensive, and technically sound study that suggests blocking the chromatin recruitment of 53BP1 using a purified inhibitory protein may facilitate gene editing for therapeutic purposes.

We thank the reviewer for this positive and supportive assessment.

Minor Points:

Some terms should be defined when first introduced in the text. For instance, "LT-HSC" (Long-Term Hematopoietic Stem Cell) is used in the main text but is first defined in the Methods section. Additionally, "MOI" should be defined upon first use.

Apologies for the omission. We have added a definition of MOI to its first use the main text (line 241-242), as well as a definition of the LT-HSC acronym to the main text (line 574-575).

Reviewer #3 (Remarks to the Author):

Perez-Bermejo JA et al. report on a very important limitation of the therapeutic application of genome editing technology: correcting genetic defects in hematopoietic stem and progenitor cells (HSPCs) without diminishing the cells' in vivo (post-transplant) long-term regenerative potential. The author describes a methodology to increase the frequency of HDR-modified alleles (therapeutic product) by specifically inhibiting the NHEJ DNA repair pathway in (HSPCs), using the CRISPR/Cas9-AAV6 platform. This study expands on an already engineered p53 inhibitor (WT to identify p53 variants that outperform the WT version in their ability to inhibit the recruitment of 53BP1 to double-strand breaks. To this end, the author developed a screening system to identify variants of i53 with higher affinity for 53BP1. Four candidates were selected and tested for their ability to increase HDR frequencies primarily at the HBB locus, though three additional genomic loci (HBA, CCR5, and IL2RG) were included.

The overall methodology used for screening and generating the i53 variants is sound. The screen was performed in HSPCs (clinically relevant cell type) and used HDR frequency as a functional readout to identify i53 variant candidates that outperform the i53 wild type.

Major concerns:

(A) While the i53 variants are more potent at inhibiting the NHEJ pathway compared to parental (WT) i53 protein and seem to disrupt only the NHEJ pathway and not MMEJ, **the increase in HDR between i53 WT and variants is incremental.**

- a. Fig. 3B i53 WT is ~%30 and i53 variants are barely reaching 40%

- b. Fig. 3A Treatment with i53 (WT or variants) at high MOI **decreases the overall % HDR compared to lower MOI. Is that due to virus-induced toxicity?**
- c. Sup Fig 3.2 D: the absolute number of HDR edited cells at the HBB locus with i53 WT vs variants is only incremental, and that's across all loci tested.
- d. Sup Fig. 3.6 A: low or high MOI no difference; 3.6B: i53 WT: 58.9 HDR events vs 60.8, 61.1, 59.4 for i53 variants

In contrast to these i53 variants (or i53 WT) Winberger, S et al., Nature Comm 2023 showed AZD7648 compound (also tested in this study) to increase the HDR frequencies from ~20% (no treatment) to ~45%. With dual AZD + PolyQi reaching close to 80% HDR. Selvaraj, S. et al., Nat. Biotech, 2023 showed that AZD7648 can achieve a drastic increase in HDR [in HSC from ~ 15% (no treatment) to almost 40% (post-treatment)], even when a suboptimal sgRNA guide was used.

We thank this reviewer for their insightful comment and concern about the improvement observed with our variants compared to i53 WT in some of the figures presented in our report, and about how the reported data compares to other recent reports (all in 2023) about DNA repair pathway inhibitors.

We were excited to see these recent studies describing additional tools (DNAPKcs and POLQ inhibition) for improving HDR-based editing outcomes and have included citations for this recent impressive work into our revised manuscript. Though we are encouraged by the drastic improvements cited by the reviewer, it is also worth contending that levels of HDR and the impact on HDR levels by any given additive is dependent on multiple factors, including cell type, donor, locus, and baseline editing levels. The latter seems very relevant in this context – we observed throughout our study that higher baseline HDR editing (e.g. as provided by increased AAV template) results in a reduced effect by the NHEJ inhibitors, presumably because there is less of a baseline NHEJ fraction to begin with. This is readily depicted in Figure 4C and matching Supplemental Figure 4.A-B: at MOI=156, where our baseline HDR% is around 15%, we observe a 2.5-fold improvement when using a 53BP1 inhibitor; in comparison we observe only a ~1.5-fold improvement at MOI=2500, where baseline HDR is 42%. (We believe this also answers the reviewer concern (b) above, about increased AAV toxicity limiting the HDR improvement).

In addition, it is important to note that in our study we provide extensive data using the same DNAPKcs small molecule inhibitor that is used in both the Wimberger et al and the Selvaraj et al studies, but applied to our HSPC, AAV-based platform. We found that the HDR improvement at the *HBB* locus was modest (and, one could argue, incremental) except when it was used in combination with 53BP1 inhibitors, although the relative improvement was locus dependent. In addition, the observation that DNAPKcs small molecule inhibitor resulted in a significant increase in off-target editing hinted at a potential pitfall of this additive for the specific application being tested in this study.

The most comparable study to ours is Selvaraj et al (2023) which uses the HBB/AAV/SNP system and does report an increase from 55% to 65% using the DNAPKi molecule on the HBB locus (an 18% increase, very similar to what we observe). The 15% to 40% increase that this reviewer refers to is observed using the CCR5 locus/gRNA, for which we use the same gRNA sequence and also see the best improvement among all loci tested when using the DNAPKi small molecule (Figure 4; raw values on fig S4.2). Looking at data from other groups we saw that the CCR5 gRNA produces a single INDEL prominently – a +1 insertion. Given our data that DNAPKcs inhibitors are more effective at targeting small insertions such as this one, it is easy to speculate that they are more effective at improving HDR at this locus compared to HBB (which has a more widespread range of INDELS of different sizes).

Regarding the POLQi compound described in Winberger, S et al., Nature Comm 2023: the reported improvement in HDR repair when used in combination with NHEJ inhibitors is notable, and agrees with our observation that combinations of inhibitors can act synergistically to improve editing outcomes (Figure 4). In our study we focus on NHEJ inhibition (53BP1 and/or DNAPKcs), and considered MMEJ inhibition to be outside the scope, but definitely acknowledge it as a potentially powerful addition to the editing toolbox. Considering the prominent MMEJ fraction observed at *HBB* in our study, we are hopeful that others will evaluate this compound relative to (and in combination with) 53BP1 or DNAPKcs inhibition at *HBB* and other clinically relevant loci in HSPCs. We are particularly eager to better understand the toxicity and safety profile of this compound and co-inhibition of MMEJ and NHEJ in HSPCs for clinical gene therapy applications. We note, however, the important considerations that reviewer #1 makes about not just increasing the HDR frequency but doing so in the context of maintaining cell number and quality. Others have found that both the DNA-PK inhibitor and PolQ inhibitors have more toxicity and detrimental effect on cell numbers (personal communication, poster abstracts). Moreover, these cellular toxic effects are amplified when combinations are used. Thus, we believe there should be an abundance of caution when perturbing DNA repair and why the use of 53BP1 inhibition seems particularly fruitful because it does not cause a loss of cell number nor cause an increase in genotoxicity.

Our goal in this report was to investigate, identify, and develop NHEJ-inhibiting additives that reliably and safely increase HDR-based outcomes in HSPCs for clinical ex vivo editing applications. We also aim to provide the community with a screening platform that can be used for the development of novel, more powerful protein-based modulators of genome editing outcomes. Ultimately, we believe that the studies referenced by this reviewer are not conflicting with our study, but rather complementary, showing how a diverse toolbox of DNA repair pathway modulators can help tackle different gene editing strategies and result in more effective control of editing outcomes. In particular, we think that the differences observed between our report and the Selvaraj et al and Wimberger et al studies are attributable to vastly different baseline HDR efficiencies, different gRNA/locus, and different nature of the HDR repair template. In particular,

Regarding the 'incremental' nature of the improvement yielded by the optimized variants, we do provide empirical data proving that the variants are significantly more potent than i53WT (both functional HDR readout and in vitro binding), potentially allowing for a reduced dose of additive and more robust outcomes. Although whether those improvements are incremental or more substantial is subjective, we would like to point out that our manuscript is not limited to the generation of these variants, but also focuses on the description of the screening pipeline used, and on the extensive characterization of editing outcomes and potential genotoxicity metrics when using these variants and/or other inhibitors. We hope that the lessons and tools provided in this report can help broaden our understanding of DNA repair inhibitors and facilitate their implementation in therapeutic editing pipelines.

We appreciate the very valuable comments of this reviewer in this section, and we have made sure to cite the aforementioned papers as well as to modify the discussion section to reflect some of the points above.

(B) Compared to recently published NHEJ (53BP1, DNAPKcs) and MMEJ (PolQ) inhibitors (Riesenbert S et al., Nat. Methods, 2022; Wimberger S et al., Nat. Comm, 2023; Selvaraj, S. et al., Nat. Biotech, 2023), the author shows that when combined with DNAPK inhibitor (AZD7648), i53 variants have locus dependent and additive effect. **A strong argument was made against using AZD7648 in the clinic due to increased off-target activity (lines 263-267).**

Was the dramatic increase (2% to 65%) in off-target activity seen across multiple loci? If not, or if they didn't test it, the author must remove this statement from the text. Alternately, the author could state that treatment with AZD during genome editing at the *HBB* locus increases off-target activity, and its clinical applications may be limited (e.g., not advised for sickle cell patients).

Though we did not test the differential impact of 53BP1 and DNAPK inhibition on off target activity at additional loci beyond *HBB* (for which we observe a ~2-fold increase), an increase in off target INDELS with AZD7648 protocols has been previously described in HSPCs for *HBB*, *HBA*, and *CCR5* off target sites (Selvaraj, S. et al., Nat. Biotech, 2023, Extended Data Fig. 6A; 2-3x increase in off-target deletions, very similar to our observation in this study).

The reviewer, however, makes a valid point on what can be concluded from our data and we have amended the text to limit our conclusions to the *HBB* locus, encouraging further studies on this end. We have also included an additional graph (Supplemental Figure 4.5) detailing the off-target editing outcomes on OT-1 when using the *HBB* gRNA. Interestingly, the increase in OT-1 editing is driven mostly by a -9 deletion, which could be associated with alternative editing pathways such as MMEJ (Selvaraj et al. also make a very similar observation in their off-target data; see discussion section).

(C) The author claims that using i53 variants increases the frequency of HDR-modified LT-HSCs and early hematopoietic progenitors while decreasing the number of cells edited by the NHEJ pathway.

- **Fig. 5A-C I can't tell if the i53 variants were better than the WT i53 (data missing).**
 - We agree with this reviewer that not including the i53WT condition in the medium/large scale editing experiment limits the interpretation of the results on the increased potency of the new variants. Running these medium scale editing experiments was very resource intensive and, beyond cost, they were limited by the amount of conditions that could be processed in parallel. This limited our ability to include full-scale control conditions. We decided to focus on the three conditions depicted (no additive mid-MOI, reduced MOI, and reduced MOI + additive) because they span the main applications that we devise for these molecules. In addition, the aim for this figure is not to provide yet another comparison of i53-WT vs variants, but to prove whether 53BP1 inhibition using these proteins is maintained at a large scale, and to provide an additional set of clinically relevant analytics. The wording in that section has been adjusted to put the focus on 53BP1 inhibitors instead of specific variants.

- **Fig 5B Editing (RNP + AAV6) adds significant toxicity to the HSPC cells. If i53 variants are superior to i53 WT and other published inhibitory compounds, then the author should have presented side by side comparison of the absolute number of colonies obtained from WT (not editing), RNP only, RNP+AAV6 (no treatment) , RNP +AAV6 (various inhibitors). Without this data it is difficult to assess the benefit (in terms of reducing toxicity) that i53 variants may have on LT-HSCs.**
 - Absolute number of colonies under those conditions are shown in Supplemental Figure 5.3A. Just like this reviewer mentions, editing did result in a significant decrease in the number of colony forming units. This panel also shows that incorporating the i53 variants had no detrimental effect in the amount of colonies. The colony counts also showed no difference in colonies when using MOI 1250 vs 625, perhaps because of the limited quantitative power of this assay (these two MOIs are too close to detect any differences). We note that in the Baik et al pre-print they report that there is an improvement in CFU number when using an MOI of 625 compared to 2500 (the MOI used in the Graphite clinical trial). While we consider that i53 variants are superior to

i53-WT, this improvement is focused on the DNA repair outcome (i.e. greatly improved HDR fraction, even beyond that of a 2x increase in AAV template). We do not claim that the i53 variants are superior in terms of safety, but found it reassuring that they did not increase toxicity in any of the metrics reported in this study.

- Fig. 5C (left panel, green bars) Same comment as for 5B. In addition, the middle bar (625 MOI + VHH variant): **with an n=2 donor and large variation, one cannot conclude. If you remove the outlier (triangle in the middle bar), all look about the same (+/- VHH; high or low MOI). The claim “significant” (line 258) is unfounded.**
 - It is our mistake for including this wording when describing conclusions of this assay - the reviewer is correct. We have removed “significantly” from the text at that line (line 492 in revised manuscript). Regarding the high variability and the use of a small sample number, note that we are reporting fold changes as they happen within donor (i.e. donor 1 condition 1 vs donor 1 condition 2 etc), which helps curve the population-level analysis (not appropriate with this *N* anyway) and the HSPC donor variability. In this regard, we have changed the way the data is displayed by showing it with a line graph that hopefully makes it easier to understand.

- The author also states (line 258) that the frequencies of HDR modification persist in **LT-HSC-derived colonies. The data in this manuscript does not support this conclusion. It takes two weeks to generate colonies, insufficient time to claim that the HDR is “persistent in the LT-HSCs.” This can only be assessed in vivo through serial engraftment studies.**
 - We thank the reviewer for pointing this out and agree that our wording was a bit careless in that regard. We have modified the wording of these statements and the figure legends wherever relevant in the text, and we have also clarified the choice of markers for this classification. This is also discussed in our response to Reviewer #1, point 12.

- While the i53 variants do not show a dramatic improvement over i53 WT, at least not in vitro, they may outperform the WT i53 or other published inhibitors in vivo. **Since most of the studies in this manuscript were performed at the HBB locus, to identify ways to improve the genome editing strategy for treating sickle cell patients, testing these variants in vivo, through xenotransplantation studies would have been necessary and in line with the scope of this study.** Given the recent publications, it would also increase the novelty of the study.
 - We agree that an *in vivo* study would have increased the impact of this manuscript and the relevance of the results. However, we consider that the focus of this report is in the introduction of a new functional screening platform for HDR-enhancing proteins in HSPCs, the structural validation of the hits identified in the screen, and the thorough analysis of editing outcomes when incorporating these variants in clinically-relevant HSPC ex vivo editing platforms, along with the comparison with the comparison to the recently described DNAPKi compound. We believe that the important contributions to the field described in this manuscript will be broadly utilized and are important to publish. While Graphite has now undergone a reverse merger and is no longer an entity to support such studies, we hope that the results reported here will motivate another group to perform the engraftment studies. We note that such studies are complex, nuanced and need to be executed in a very thoughtful manner as type of mouse, quality of reagents, MOI of AAV, cell dose all can affect the results and complicate interpretations of any results and the impact of a high-affinity i53 peptide.

- Reducing the time of ex vivo culture and manipulation (from 5 days to 4 or 3) is another critical parameter in preserving the cells' engraftment potential. **An additional interesting experiment would be to see if treating CD34+ at day 2 (rather than day 3) of ex vivo expansion increases the frequency of HDR by day 3 or, at most, day 4 (instead of day 5).** If it does, how would these variants compare to other published inhibitors? NHEJ is active throughout cell cycle stages, while HDR machinery is only active at the S-G2 phases. If the cells are treated on day 2 (the time when the cells complete the G1 phase and enter the S phase) and allowed to cycle for only an additional day, would a high HDR frequency be observed while increasing the cells' regenerative potential (CFU and engraftment)?
 - We thank this reviewer for the excellent suggestion. We agree that studying the intersection of DNA repair modulation and cell culture/editing timing would be very interesting, and worthy of a full-scale manuscript in itself. In this manuscript, we focused on the single variable of inclusion of a newly developed peptide inhibitor of 53BP1 using a baseline HDR-editing process. As noted in the first sentence, the development of a potentially shorter and improved culture conditions would be very interesting but also would involve investigating a number of intersecting variables including cytokine mix, type of culture media, duration of culturing pre and post-electroporation, the electroporation process itself (or even the use of a non-electroporation based delivery methods), the use of bioreactors, the potential use of other additives to minimize the stress of ex vivo culturing of CD34 cells and genetic engineering... The list is long and highlights that there remain numerous other avenues of investigation to continue to iterate and improve HDR-editing in CD34+ human HSPCs and other cell types. Given the recent discussion of costs of gene therapy clinical costs, areas of study in the future should also include not just increasing the frequency of HDR and cell number and quality but also methods to decrease costs and thereby potentially increase accessibility. Again, those go beyond the scope of this work though we note that by using the peptide to decrease the dose of AAV6 by 4-fold or more may be one step in that direction since the peptide is used at lower doses and is easy to manufacture from bacteria. Nonetheless, we are hopeful that by publishing this work, others may utilize these new variants (alongside, as the reviewer mentioned, other HDR-boosting additives, such as DNAPKi and POLQi) to probe the impact of DNA repair pathway inhibition in the context of a reduced time for ex vivo culture and manipulation (and the impact this may have on HDR levels post-*in vivo* engraftment).

REVIEWER COMMENTS

Reviewer #1 (Remarks to the Author):

The revised manuscript focuses in on the major innovations, namely a screening platform in HSCs and the resultant novel i53 variants that seem to be further suppress NHEJ editing to allow for improved ratio of successful HDR in cells left at the end of the editing protocol. The authors in their rebuttal indicate that they have addressed the concerns about cell number data being left out or otherwise obfuscated by the data as presented in their main and supplementary figures. The final edited cell yield is of equivalent or perhaps greater importance to cell therapy solutions as % edited cells or total cells at 24 hours, yet the authors fail to disclose edited cell yields in any of their updated figures or text. This is really important data to include if the claim is novel i53 variants that could be used for cell therapy. If the claim is improving cell editing (not therapy), then even if cell yield is decreased for certain situations the fraction of HDR may be more useful than the total number of cells. Overall the manuscript is impressive and should be of high appeal to Nature Communications readership and the cell genome engineering field more broadly. No additional data are required if the claims are adjusted to make clear they do not have data supporting improved edited cell yields when using i53 variants compared to without. Or, the manuscript could be enhanced by adding greater details into edited cell yields at timepoints beyond 24 hours.

Specific Issues:

Lines 26-27: "When applied at manufacturing scale, the incorporation of improved variants results in a significant increase in cells with at least one repaired allele"

- This is ambiguous. Given the paucity of cell yield data provided in the revised manuscript, would be better to state something like 'significant increase in fraction of cells with at least one repaired allele'.

Line 150: "The inclusion of the i53 variants resulted in equivalent cell numbers and cell viability, and no added toxicity, as the control samples with no protein (Supplementary Fig. S3.5).

- The flow cytometry based data does not indicate anything about cell numbers. If cells died between time of editing and flow cytometry, and were subsequently washed away, they would not be assessed by these endpoint flow assays. Similarly if cells proliferated, they would be indistinguishable from cells that did not proliferate as much or at all. This could be a starting population of 250k, final population of 10k or 1000k, with the same flow plots but vastly different interpretations of cell viability and toxicity. To illustrate the point, imagine a starting point of 100 cells that cannot divide, then a treatment added into the cell mixture that causes 90 cells to explode but doesn't touch the remaining 10 (for example, a laser). If you did flow cytometry on the population of cells two days later, those 10 cells would be all that remains, and the analysis would look like 10 out of 10 would appear alive by Live/Dead stain, yet the laser treatment would be highly toxic (10% viability or 10% cell yield). If you activated those cells and suddenly they can divide every 24 hours, those 10 cells could become ~40 cells by day two, and flow would again show 40 out of 40 cells (!005) would appear alive by Live/Dead stain, yet the relative cell yield would appear to be 40%.

- Again, absolute cell numbers (starting # of cells, final count of cells) would be the ideal data to include here. If not available, relative cell numbers (fold change of final cell count compared to initial cell count). If neither available, or if the cell count was not measured at the end of the experiment, then please remove the claim of equivalent cell numbers and viability and no added toxicity to be congruent with the data.

Supplemental Figure S4.4

- This is the first figure updated to include some data on total cell recovery, however it is only presented at 24 hours. L67R appears to have increased P21 and H2AX, but additional more conventional markers are not shown. Further, the most important metric is not how many cells are present at 24 hours but how well the cells can grow post-editing, which many other papers have already shown that RNP electroporation alone inhibits cell growth compared to untreated. That the authors observe similar numbers of cells 24 hours post treatment compared to untreated control only highlights the fact that they simply did not wait long enough to make the comparison – which is very

strange considering that the methods indicate they did sequencing at 2 days post editing and protein level (FACS) assays at 3-5 days post editing. In all conditions except for MOI 625 the trend appears to be L67R with lower cell counts than untreated. The authors do not report comprehensive cell yield outcomes at either 2, 3, or 4 days post editing, which is when global toxicity of electroporation, +/- AAV6, +/- i53 protein/variant would be most expected to show through.

- If the authors have data for cell count or some other measure of cell yield beyond 24 hours, this would be most helpful for readers and the field.

- Was there no need to recount the cells at least one time to normalize concentrations, or if not, were no additional cytokines or media added between days 1 and 3-5 post editing?

Line 228-229, 231 and S5.1E: "Similarly, we observed no difference in cell recovery (Supplementary Fig. S5.1E), when i53 variants were included in the editing protocol".

- The data in Fig S5E are total cell yield data (not edited cell yield), without indication as to which of those cells were actually edited (which would be difficult to define without a protein level readout, and SNP KI here is not readily adaptable to a protein level readout without destroying the individual cells). This is the only data in the entire paper that directly shows no worse toxicity of the i53 variant VHH in the context of electroporation + AAV6 (both of which are known to reduce cell yield compared to unedited).

- There is no indication of the timepoint here. If this is 24 hours as in Fig S4.4, then the same criticism is valid that they simply did not wait long enough to count their cells. It seems unlikely that the authors would edit such a large scale of cells then only count at 24 hours. Again, this raises concerns that the authors are hiding data that is counter to their intended claims.

- There is neither an unedited control nor a control of electroporation with RNP only, both of which are of course reasonable to leave out at this large scale. However there are no additional pilot or small scale data (other than the problematic Fig S3.5 and S4.4) to support a claim of minimal global toxicity.

- The implication is that it is still possible (and there is no data given to the contrary) that the i53 variants can be more obviously toxic after many days of cell culture (or transplantation), or more toxic in other contexts (such as completely nonviral Cas9 plus HDRT both via electroporation).

- Thus, this claim is again ambiguous. Would be better stated as 'we observed no difference in cell recovery at [24 hours] ... when i53 variants were included in an editing protocol based upon electroporation and AAV6 HDRT delivery.'

Reviewer #2 (Remarks to the Author):

The authors have appropriately addressed all of my comments. I highly recommend the publication of this manuscript in Nature Communications.

Reviewer #3 (Remarks to the Author):

I thank the author for addressing all my points.

I request that the author include graphs showing the absolute numbers of cells before and after treatment. Too often, they present "fold over no treatment," making it difficult to assess viability and how many cells were modified. It makes it easier for a reviewer to assess the full effect of the additive – for example, Figure 4C. This data should not be buried in supplementary figures.

The field is aware of variability between HSPC donors (possibly an epigenetic factor not yet investigated). The AAV platform used in this study and that in Sevaraj should be the same, if not identical. If not, how does it differ that might affect the HDR outcome?

Perez-Bermejo et al.'s study and Selvaraj et al.'s testing of different small molecules showed an incremental increase at the HBB locus. Since the author is primarily interested in boosting the HDR at

the HBB locus (without increasing AAV MOI), would their newly identified i53 compounds be sufficient to make a difference in clinical settings? What is the threshold of correction at the HBB locus (frequency of HDR at the HBB locus) that would be necessary to achieve correction? And has this threshold not yet been achieved or surpassed?

The author states, " Our goal in this report was to investigate, identify, and develop NHEJ-inhibiting additives that reliably and safely increase HDR-based outcomes in HSPCs for clinical ex vivo editing applications." The levels of HDR at the HBB locus already published demonstrate that high levels of genome editing (HDR) are possible and attainable. The question that still remains and is not addressed here is how to prevent hematopoietic stress HDR in LT-HSCs and ensure the regenerative properties of these cells are preserved in the clinic.

The author states: "We believe that the studies referenced by this reviewer do not conflict with our study." It's not conflicting; it's just not much improvement from what was published.

The author states: "We agree with this reviewer that not including the i53WT condition in the medium/large scale editing experiment limits the interpretation of the results on the increased potency of the new variants. Running these medium scale editing experiments was very resource intensive and, beyond cost, they were limited by the amount of conditions that could be processed in parallel. This limited our ability to include full- scale control conditions." Running a small-scale editing experiment first, which includes all the proper controls, would be informative and less expensive.

Figure 5A: Please add a third biological replicate for the conditions where it's missing (Top: 625 MOI, 625 VHH), Bottom: 625 MOI, and 1250. Determine then if there are still statistical differences.

The author states regarding Figure 5C "Regarding the high variability and the use of a small sample number, note that we are reporting fold changes as they happen within donor (i.e. donor 1 condition 1 vs donor 1 condition 2 etc), which helps curve the population- level analysis (not appropriate with this N anyway) and the HSPC donor variability. In this regard, we have changed how the data is displayed by showing it with a line graph that hopefully makes it easier to understand." Please include absolute numbers, too: the numbers of HSPCs at the start of the experiment and those left post-treatment that have successfully undergone HDR.

RESPONSE TO COMMENTS FOR RESUBMISSION

PEREZ-BERMEJO ET AL, 2024 (NATURE COMMUNICATIONS NCOMMS-23-36442-A and NCOMMS-23-36442-T)

We thank the editor and all the reviewers for carefully reading and reviewing our revised manuscript. We very much appreciate the reviewers' continued efforts in helping us refine our manuscript and value the opportunity to address the reviewers' remaining concerns.

REVIEWER COMMENTS

Reviewer #1 (Remarks to the Author):

The revised manuscript focuses in on the major innovations, namely a screening platform in HSCs and the resultant novel i53 variants that seem to be further suppress NHEJ editing to allow for improved ratio of successful HDR in cells left at the end of the editing protocol. The authors in their rebuttal indicate that they have addressed the concerns about cell number data being left out or otherwise obfuscated by the data as presented in their main and supplementary figures. The final edited cell yield is of equivalent or perhaps greater importance to cell therapy solutions as % edited cells or total cells at 24 hours, yet the authors fail to disclose edited cell yields in any of their updated figures or text. This is really important data to include if the claim is novel i53 variants that could be used for cell therapy. If the claim is improving cell editing (not therapy), then even if cell yield is decreased for certain situations the fraction of HDR may be more useful than the total number of cells. Overall the manuscript is impressive and should be of high appeal to Nature Communications readership and the cell genome engineering field more broadly. No additional data are required if the claims are adjusted to make clear they do not have data supporting improved edited cell yields when using i53 variants compared to without. Or, the manuscript could be enhanced by adding greater details into edited cell yields at timepoints beyond 24 hours.

Thank you for the insightful comment and suggestion. We agree that edited cell yields are important to report, as they are crucial to the utility and applicability of the protein reagents to gene therapy protocols.

In our study, we started by focusing primarily on assessing impact of protein variants on editing outcomes (quantified as HDR-corrected alleles or cells %, analyzed by NGS or flow cytometry) as a way to validate potency and to prioritize candidates. Though we were not explicitly quantifying toxicity readouts at early stages of candidate prioritization, we did not observe any obvious negative impacts when these proteins were expressed in or introduced via nucleofection to the HSPCs. Once we narrowed down our selection of candidates, we began to look at % apoptotic cells (by flow cytometry for a rough identification of toxicity liabilities in a dose-dependent manner, as reported in Fig S3.5). Once we settled on candidate proteins and working concentrations, we began quantifying relevant cell health metrics, and systematically recording cell yield numbers. These are shown accordingly in the experiments in Figures 4 and 5, but are especially important for the latter since that's the experiment where editing was performed closest to at-scale (and actually employing all instrumentation/steps of the clinical implementation). We didn't observe an impact of the i53 variants on cell yields in either of these experiments (a small nonsignificant dip is visible in panel 5B but it appears to be driven by only 1 of the donors, and apparently due to a loss during zapping, not in post-culture incubation – we have noted it in-text).

We have revisited our experiment logs and cross-checked all the data to provide more explicit information about cell numbers and yield (recovery at 24h compared to pre-electroporation). This data is now presented in the supplements for Figure 4 and 5 (supplemental figures S4.4C and S5.1E), panel B of main Figure 5, and supplemental table S5.1 (which contains all the cell numbers through the process of editing in large scale). Since our readout of corrective editing outcomes in these experiments is NGS, we are thus limited to quantifying %HDR-corrected alleles in the bulk edited cell populations and we cannot provide absolute edited/corrected cell numbers (i.e. # of cells with at least one corrected allele). However, we hope that the combination of cell counts and %edited alleles can provide a reasonable estimate of “number of edited cells” at harvest (24 h). We have made sure to carefully revise the text and only mention ‘increase in edited

cells' where cell numbers could be provided, and just refer to "fraction of edited cells/alleles" everywhere else.

Regarding the study of yields/toxicity beyond 24h, we definitely agree with the reviewer that such data would have been a great addition to this manuscript. Although initial experiments were performed with readouts at 48h and 5d post-editing for NGS and %GFP, respectively, the later shift to a 24h timepoint is also related to the pivot towards using a more clinically relevant protocol - the procedure approved to be used by Graphite Bio required a 24h post-editing cryopreservation step, so we focused on editing percentages at that timepoint ('drug product'). Again, we agree that future studies should look into cell behavior beyond that timepoint. We understand this is a limitation of our study and have made sure to change the main text to mention it, and to limit claims to the 24h timepoint.

Specific Issues:

Lines 26-27: "When applied at manufacturing scale, the incorporation of improved variants results in a significant increase in cells with at least one repaired allele"

- This is ambiguous. Given the paucity of cell yield data provided in the revised manuscript, would be better to state something like 'significant increase in fraction of cells with at least one repaired allele'.

Thanks for this valid feedback. As we mentioned above, given that we provide both cell yields (at 24 h) and % colonies/alleles edited, and that cell yields are not significantly altered upon incorporating the i53 variant, we can estimate the number of edited cells to also be improved. We have anyway updated the abstract sentence to mention 'fraction of cells', to remain conservative in our claims.

Line 150: "The inclusion of the i53 variants resulted in equivalent cell numbers and cell viability, and no added toxicity, as the control samples with no protein (Supplementary Fig. S3.5).

- The flow cytometry based data does not indicate anything about cell numbers. If cells died between time of editing and flow cytometry, and were subsequently washed away, they would not be assessed by these endpoint flow assays. Similarly if cells proliferated, they would be indistinguishable from cells that did not proliferate as much or at all. This could be a starting population of 250k, final population of 10k or 1000k, with the same flow plots but vastly different interpretations of cell viability and toxicity. To illustrate the point, imagine a starting point of 100 cells that cannot divide, then a treatment added into the cell mixture that causes 90 cells to explode but doesn't touch the remaining 10 (for example, a laser). If you did flow cytometry on the population of cells two days later, those 10 cells would be all that remains, and the analysis would look like 10 out of 10 would appear alive by Live/Dead stain, yet the laser treatment would be highly toxic (10% viability or 10% cell yield). If you activated those cells and suddenly they can divide every 24 hours, those 10 cells could become ~40 cells by day two, and flow would again show 40 out of 40 cells (100%) would appear alive by Live/Dead stain, yet the relative cell yield would appear to be 40%.

- Again, absolute cell numbers (starting # of cells, final count of cells) would be the ideal data to include here. If not available, relative cell numbers (fold change of final cell count compared to initial cell count). If neither available, or if the cell count was not measured at the end of the experiment, then please remove the claim of equivalent cell numbers and viability and no added toxicity to be congruent with the data.

Thank you for the explanation and the remarks. We agree that the data does not support a claim such as that one. No cell counts were systematically taken at this stage (as described above), and only a flow cytometry apoptosis assay was used. We have replaced this statement with a more appropriate one.

Supplemental Figure S4.4

- This is the first figure updated to include some data on total cell recovery, however it is only presented at 24 hours. L67R appears to have increased P21 and H2AX, but additional more conventional markers are not shown. Further, the most important metric is not how many cells are present at 24 hours but how well the cells can grow post-editing, which many other papers have already shown that RNP electroporation alone inhibits cell growth compared to untreated. That the authors observe similar numbers of cells 24

hours post treatment compared to untreated control only highlights the fact that they simply did not wait long enough to make the comparison – which is very strange considering that the methods indicate they did sequencing at 2 days post editing and protein level (FACS) assays at 3-5 days post editing. In all conditions except for MOI 625 the trend appears to be L67R with lower cell counts than untreated. The authors do not report comprehensive cell yield outcomes at either 2, 3, or 4 days post editing, which is when global toxicity of electroporation, +/- AAV6, +/- i53 protein/variant would be most expected to show through.

Thanks for the feedback on this dataset. As this reviewer points out, we do provide cell counts/yield at harvest, but these are limited to 24h post-editing (to match Graphite Bio's approved clinical protocol; see our first response above for details). We have adjusted the text so the claims are limited to 24h.

We have also adjusted the methods, as well as the figure legends to more clearly convey when a 24 h NGS readout was taken – we apologize that the text was confusing as to when endpoint readouts were taken, and seemed to suggest that cell viabilities were taken at an earlier timepoint than NGS.

Lastly, the slightly reduced cell counts when using L67R this reviewer observes did not flag as significant in a relatively high-power statistical analysis (main effects in a two-way ANOVA), so we decided not to report it. Upon reviewing all the ratios manually, it looks like some of the MOIs (mostly the lower ones) had ~10% reduction in yields for all donors studied, whereas the other MOI conditions had equal or increased yields. Given that the effect is not consistent and actually within error for the cell counting, we have kept it as-is. We have however included a note on the need of further studies in this regard, both in the results section and the discussion section.

- If the authors have data for cell count or some other measure of cell yield beyond 24 hours, this would be most helpful for readers and the field.

- Was there no need to recount the cells at least one time to normalize concentrations, or if not, were no additional cytokines or media added between days 1 and 3-5 post editing?

Unfortunately, due to the reasons outlined above (see the beginning of response to this reviewer) we did not systematically collect cell counts beyond 24h in any of our experiments. This is a caveat of the study that we have made sure to highlight in the discussion section.

In those experiments where the cells were kept up to day 5 post editing (i.e. %GFP knock-in readouts) cells were not perturbed until endpoint, except for AAV washout at 24h and no cell counts were recorded then (we had previously observed a consistent cell loss across conditions, and often cell samples were too small to take out a substantial aliquot for counting).

Line 228-229, 231 and S5.1E: “Similarly, we observed no difference in cell recovery (Supplementary Fig. S5.1E), when i53 variants were included in the editing protocol”.

- The data in Fig S5E are total cell yield data (not edited cell yield), without indication as to which of those cells were actually edited (which would be difficult to define without a protein level readout, and SNP KI here is not readily adaptable to a protein level readout without destroying the individual cells). This is the only data in the entire paper that directly shows no worse toxicity of the i53 variant VHH in the context of electroporation + AAV6 (both of which are known to reduce cell yield compared to unedited).

- There is no indication of the timepoint here. If this is 24 hours as in Fig S4.4, then the same criticism is valid that they simply did not wait long enough to count their cells. It seems unlikely that the authors would edit such a large scale of cells then only count at 24 hours. Again, this raises concerns that the authors are hiding data that is counter to their intended claims.

Although we understand this reviewer's concerns, we can confirm that we are showing all the data we have produced for this experiment, and are hiding none - the cells were cryopreserved at 24 h (mirroring our clinical editing protocol) to enable downstream studies, specifically an in vivo transplantation study in mice which requires a high number of cells. Unfortunately, these studies were halted when the sickle cell program

was terminated at Graphite mid 2023. Though we were not able to complete these types of long-term viability and toxicity studies at Graphite, we are hopeful that by sharing our results we encourage other studies that continue to further assess the viability of the implementation of these additives in clinical protocols. This point has been made clear in the discussion. We have also included cell yields from this experiment into the main figure (5B), and cell counts pre-editing, post-editing and at harvest (24h) in supplemental figure S5E. Supplemental Table 5.1 also includes all cell counts at those timepoints.

As this reviewer points out, it is not possible for us to obtain a direct quantitation of the amount of cells edited instead of a relative % of edited cells or alleles, but we hope that by providing both ratios and cell yield numbers we can provide a good overall picture of the editing of cells (at 24h).

- There is neither an unedited control nor a control of electroporation with RNP only, both of which are of course reasonable to leave out at this large scale. However there are no additional pilot or small scale data (other than the problematic Fig S3.5 and S4.4) to support a claim of minimal global toxicity.

We had saved a fraction of cells from the large (medium) scale editing experiments for Untreated (~40M cells) and RNPonly (80M cells), which we initially thought would not be necessary to include but that we now include as valuable controls since they were processed in parallel and using the same protocol than the experimental conditions shown in Figure 5. We have included editing efficiencies in the panel 5A, and the cell yields in panel 5B, along with related supplemental figures and panels. When taken altogether, we observe a reduced yield in all electroporated conditions compared to Untreated, and an additional reduction in yield in those conditions containing AAV template. We do not observe a significant dip in yield or cell number in the conditions containing i53 variant inhibitor (although, as mentioned at the beginning of this response, there is a reduction in yield for one of the donors).

- The implication is that it is still possible (and there is no data given to the contrary) that the i53 variants can be more obviously toxic after many days of cell culture (or transplantation), or more toxic in other contexts (such as completely nonviral Cas9 plus HDRT both via electroporation).

- Thus, this claim is again ambiguous. Would be better stated as 'we observed no difference in cell recovery at [24 hours] ... when i53 variants were included in an editing protocol based upon electroporation and AAV6 HDRT delivery.'

Thanks for the suggestion, which we have added to our text.

Overall, we believe that our data supports that i53 variants are not obviously toxic at 24h. Notably, these additives were actually being considered as an addition to the clinical editing protocol at Graphite Bio, had they been through more extensive characterization particularly in an engraftment study. However, we agree with this reviewer that further studies on later timepoints will be of importance to fully characterize the applicability of these variants and 53BP1 inhibitors overall.

Reviewer #2 (Remarks to the Author):

The authors have appropriately addressed all of my comments. I highly recommend the publication of this manuscript in Nature Communications.

Thank you for the valuable comments throughout the review process.

Reviewer #3 (Remarks to the Author):

I thank the author for addressing all my points.

I request that the author include graphs showing the absolute numbers of cells before and after treatment. Too often, they present “fold over no treatment,” making it difficult to assess viability and how many cells were modified. It makes it easier for a reviewer to assess the full effect of the additive – for example, Figure 4C. This data should not be buried in supplementary figures.

Thanks for the suggestion. We agree that cell counts are important to get a full picture on the usefulness of these molecules. As mentioned at the beginning of our response to reviewer 1, roughly the first half of the editing experiments showed in this manuscript were done in small scales and/or using GFP readout with the aim of assessing potency, and cell counts were not rigorously recorded. Fortunately, we started recording cell counts as we ramped up towards the implementation of these additives in our clinical editing protocol, as shown for experiments in Figure 4C/D and, most importantly, Figure 5.

We have modified Figure 5 to include a ‘yields’ panel, as well as full counts on Supplemental Figure 5.1E and Supplemental Table 5.1. We have also included cell counts for the experiment on Figure 4C, which this reviewer directly mentions (we used 1M cells as input so the trends in cell counts match the trends in yields).

The field is aware of variability between HSPC donors (possibly an epigenetic factor not yet investigated). The AAV platform used in this study and that in Selvaraj should be the same, if not identical. If not, how does it differ that might affect the HDR outcome?

It is correct that the AAV platform used in this study is very similar to those used in Selvaraj et al. We also agree that there are differences between the HDR outcomes reported here and those described in the Selvaraj report. The Selvaraj report presents HDR outcomes in many cell types in addition to donor-derived HSPCs (including PSC cell lines, T cells, B cells, and HBECs) and there are differences in analysis methodology and timing. For example, in the Selvaraj report, ICE and ddPCR was used to quantify allelic distribution post editing; in our study we leveraged NGS to provide a more accurate readout and for key decision making, including as an FDA-approved qualifying assay for the clinical product at Graphite Bio, despite higher cost and turnaround times.

Despite these differences, we believe that our overall results regarding DNAPKi efficiencies in HSPCs are not dissimilar to what is reported in Selvaraj et al when the same gRNAs are used. When Selvaraj et al report editing of the HBB locus in HSPCs (using the same gRNA used in most of the experiments in our study) they observe a modest improvement that is similar to what we observe - see Figure 2D (~15% improvement, as determined by ICE analysis) and Extended Data Figures 6C (~12% improvement, as determined by ICE analysis) and 8D (~18-22%, depending on MOI, as determined by ICE analysis) on their manuscript, versus a ~20-25% improvement in our NGS figures S4.1, S4.4B, and 4.4C.

The majority of experiments described in Selvaraj et al are using a gRNA targeting the CCR5 locus (see Fig 2A for data surrounding this locus in HSPCs, as analyzed by ddPCR and ICE analysis, ~2X improvement in %HDR), which we also used in Figure 4A (using a flow based analysis, ~1.6X improvement in GFP+ expressing cells). Across the loci tested, we observed CCR5 to be where DNAPKi provided the best improvement (possibly because the INDEL pattern mostly consists of a +1 insertion, which potentially gets preferentially inhibited by DNAPKi, as we observe in Figure 4B).

Overall, we think these observations support the notion that different editing additives (such as DNAPKi or i53 and variants) can be used in a locus-dependent manner to achieve best results, and that studies like the one we present here are important to help expanding the genome editing toolbox.

Perez-Bermejo et al.'s study and Selvaraj et al.'s testing of different small molecules showed an incremental increase at the HBB locus. Since the author is primarily interested in boosting the HDR at the HBB locus (without increasing AAV MOI), would their newly identified i53 compounds be sufficient to make a difference in clinical settings? What is the threshold of correction at the HBB locus (frequency of HDR at the HBB locus) that would be necessary to achieve correction? And has this threshold not yet been achieved or surpassed?

Thanks for the thoughtful comment. In our manuscript, we present on the optimization and implementation (with relatively extensive characterization) of a completely different family of inhibitors targeting the 53BP1 enzyme. Notably, these inhibitors are protein-based rather than small molecules, which in the specific case of delivery by nucleofection results in easy delivery with reduced half-life and potentially higher specificity. Although we studied the use of both inhibitors in combination (Fig 4A-B), the combination presented some predicted liabilities (i.e. toxicity; increase in off-target editing; difficulty to control residence time), and it was not dramatically better for the HBB locus editing in particular. We do believe that 53BP1 inhibitors, and in particular i53 optimized variants, have the potential to impact clinical applications of gene therapy. However, the choice of inhibitor needs to be evaluated in a case- and locus- specific manner, as we outlined in our response above this one (e.g. both Selvaraj et al and us observed that DNAPKi is much more effective on the CCR5 locus editing strategy).

Regarding the threshold for correction, it is widely believed that a correction of 15% of cells / 30% alleles is sufficient to achieve correction of sickle cell disease. Although baseline editing efficiencies *in vitro* are definitely above that level (see our study and others), there is a major concern that engraftment would result in an enrichment of cells with INDELS, dramatically reducing the fraction of long-term engrafted cells that are HDR-corrected (this has been described by others, for example on Shin et al, 2020, and mentioned by us in the main text). Indeed, it is now public information that this factor partially contributed to the negative results in Graphite Bio's clinical trial. The i53 variants program we present in our manuscript was actually developed as a contingency plan in case these issues arose, and we now share our results with the rest of the scientific community to help in the development of safer gene therapies with more predictable outcomes.

The author states, "Our goal in this report was to investigate, identify, and develop NHEJ-inhibiting additives that reliably and safely increase HDR-based outcomes in HSPCs for clinical ex vivo editing applications." The levels of HDR at the HBB locus already published demonstrate that high levels of genome editing (HDR) are possible and attainable. The question that still remains and is not addressed here is how to prevent hematopoietic stress HDR in LT-HSCs and ensure the regenerative properties of these cells are preserved in the clinic.

We wholeheartedly agree with this reviewer that the long-term potential for engraftment of edited cells (and whether the additives improve it) is a critical factor to evaluate the success of a prospective gene therapy. As we mentioned in our response to reviewer 1, the cells from medium-scale editing (Figure 5) were originally intended to be used in a mouse engraftment study, but the program was discontinued before it could be completed. We performed the "phenotypic LTHSC" panel (Figure 5D) as the closest we could get to engraftment information, but it definitely does not replace it. Future studies will have to address improvement over engrafted cells. We have modified the discussion in this manuscript to highlight this limitation even more clearly.

The author states: "We believe that the studies referenced by this reviewer do not conflict with our study." It's not conflicting; it's just not much improvement from what was published.

We appreciate the concerns regarding the novelty of our manuscript. As mentioned above, in this study we are presenting the optimization and implementation of a separate family of NHEJ inhibitors which seem to display a different profile of INDEL inhibition compared to DNAPKi, and would be more useful in a separate set of loci. This could also be relevant should any important liabilities arise from the use of DNAPKi in gene therapy protocols, or in case the protein-based inhibitors are preferred to small molecules. Beyond that, we also introduce a screening method that utilizes a relevant gene editing protocol to identify and refine protein-based gene editing reagents beyond the i53 variants. Lastly, we also present the implementation of the novel i53 variants in a medium-scale protocol that mirrors the clinical editing protocol almost identically, which provides an additional level into the feasibility of incorporating these additives in actual gene therapy programs.

The author states: "We agree with this reviewer that not including the i53WT condition in the medium/large scale editing experiment limits the interpretation of the results on the increased potency of the new variants. Running these medium scale editing experiments was very resource intensive and, beyond cost, they were

limited by the amount of conditions that could be processed in parallel. This limited our ability to include full-scale control conditions." Running a small-scale editing experiment first, which includes all the proper controls, would be informative and less expensive.

We have reviewed Figure 5 and associated supplemental data to now include two controls – an Untreated condition and an 'RNP only' (no AAV) condition. These controls used slightly lower cell numbers (see data) but were processed in parallel to the other experimental conditions. Sadly, we did not manage to include a i53WT due to the reasons we outlined in our previous response (an additional condition including additive was particularly hard to include due to the added step of mixing in the i53 variant).

We appreciate the suggestion of running a small-scale experiment first. Figure 3 and all associated supplements showcase small-scale editing experiments where we could incorporate the aforementioned controls, and which we leveraged extensively to refine the conditions we then tested in the medium/large scale experiment. Of note, due to the vessel size over all steps - cell culture bags, nucleofection chamber, closed-system fluidics system - we could not simply do the large scale process with lower cell numbers, and we would still be limited in how many conditions we can process in parallel.

Figure 5A: Please add a third biological replicate for the conditions where it's missing (Top: 625 MOI, 625 VHH), Bottom: 625 MOI, and 1250. Determine then if there are still statistical differences.

We agree that adding a third biological replicate (donor) would increase our confidence in the observed results. Unfortunately, the team in charge of this analysis only allocated resources for the analysis of two of the donors. The panel is very comprehensive and the most important population (phenotypic LT-HSCs) is low in abundance, which means it took multiple days to perform. This limited how many samples could be analyzed. Sadly, we cannot add a third replicate now due to the Sickle Cell program at Graphite having been discontinued and the laboratories, dismantled.

We have updated the main text to explicitly mention we only analyzed two out of the three donors in this experiment, and also modified the discussion section to suggest the need for studying a broader range of donors.

The author states regarding Figure 5C "Regarding the high variability and the use of a small sample number, note that we are reporting fold changes as they happen within donor (i.e. donor 1 condition 1 vs donor 1 condition 2 etc), which helps curve the population-level analysis (not appropriate with this N anyway) and the HSPC donor variability. In this regard, we have changed how the data is displayed by showing it with a line graph that hopefully makes it easier to understand." Please include absolute numbers, too: the numbers of HSPCs at the start of the experiment and those left post-treatment that have successfully undergone HDR.

Thanks for the suggestion. As mentioned above, we have updated the manuscript to include extensive data on cell numbers for the large scale experiments: pre-nucleofection, post-nucleofection, and at harvest (24h). See Supplemental Table 5.1 and also Figure 5B and Supplemental Figure 5.1B.

REVIEWERS' COMMENTS

Reviewer #1 (Remarks to the Author):

Appreciate the extensive thought process and clarity given in the rebuttal. The changes made greatly improve the manuscript, including by clarifying that the toxicity statements are limited to 24 hours, as that is all the data available. The authors have addressed my concerns. The newly revised manuscript is impressive and should be of high appeal to Nature Communications readership and the cell genome engineering field more broadly. I recommend publication.

Reviewer #3 (Remarks to the Author):

I thank the author for carefully addressing my concerns. I have recommended the manuscript for publication.